# Curved adhesions mediate cell attachment to soft matrix fibres in three dimensions

Wei Zhang[1,2], Chih-Hao Lu[1,2], Melissa L. Nakamoto [1,2], Ching-Ting Tsai[1,2], Anish R. Roy [1], Christina E. Lee[2,3], Yang Yang[1,2], Zeinab Jahed[1,2,4], Xiao Li[1,5] & Bianxiao Cui [1,2] ✉

Integrin-mediated focal adhesions are the primary architectures that transmit forces between the extracellular matrix (ECM) and the actin cytoskeleton. Although focal adhesions are abundant on rigid and flat substrates that support high mechanical tensions, they are sparse in soft three-dimensional (3D) environments. Here we report curvature-dependent integrin-mediated adhesions called curved adhesions. Their formation is regulated by the membrane curvatures imposed by the topography of ECM protein fibres. Curved adhesions are mediated by integrin αvβ5 and are molecularly distinct from focal adhesions and clathrin lattices. The molecular mechanism involves a previously unknown interaction between integrin β5 and a curvature-sensing protein, FCHo2. We find that curved adhesions are prevalent in physiological conditions, and disruption of curved adhesions inhibits the migration of some cancer cell lines in 3D fibre matrices. These findings provide a mechanism for cell anchorage to natural protein fibres and suggest that curved adhesions may serve as a potential therapeutic target.

Mammalian cells adhere to the ECM through integrin-mediated adhesions to transmit environmental cues and regulate their behaviours in physiology and pathology[1,2]. Focal adhesions and related structures have been identified as the primary architectures that connect the ECM to the actin cytoskeleton and transmit mechanical forces. The activation of integrin receptors in focal adhesions requires the engagement of force-sensitive intracellular proteins[2], especially talin[3,4]. This force-dependent process is a key regulatory step in the formation of focal adhesions[5], making them sensitive to ECM stiffness and cell contractility[6–8].

Focal adhesions are prominent on two-dimensional (2D) rigid substrates. However, they are much smaller in size[9], barely discernible[10] or not observed[11,12] in soft 3D matrices. This is partly because talin activation in focal adhesions requires a minimum substrate rigidity of 5 kPa[13], which is higher than the rigidity of most ECM protein fibres[14]. Cell adhesions in three dimensions depend on various factors, such as ECM composition, density and organization[15]. Despite the complexity, force-dependent integrin activation is thought to be the key step for cell adhesions in 3D structures[16]. Interestingly, in conditions in which focal adhesions are not observed, adhesion proteins like talin still play a role in cell motility in 3D structures[11], which indicates that alternative adhesion architectures form on soft 3D fibres.

Recent studies have identified clathrin-containing integrin adhesions[17], including flat clathrin lattices[18,19], tubular clathrin–AP2 lattices[20] and reticular adhesions[21]. It is possible that clathrin lattices may participate in cell adhesions to soft fibres. However, clathrin-containing adhesions are generally devoid of key mechanotransduction components such as actin and talin[17].

A key geometrical feature of ECM fibres is their cylindrical and curved surface. It has long been proposed that this fibre geometry may play an important role in cell–matrix adhesions by inducing plasma membrane curvature[22]. However, a mechanism connecting membrane curvature and integrin adhesion has yet to be established. In this study, we report the identification of curved adhesion. The formation

[1]Department of Chemistry, Stanford University, Stanford, CA, USA. [2]Wu-Tsai Neuroscience Institute and ChEM-H institute, Stanford University, Stanford, CA, USA. [3]Biophysics Program, Stanford University School of Medicine, Stanford, CA, USA. [4]Present address: Department of Nanoengineering, University of California, San Diego, CA, USA. [5]Present address: School of Mechanical Engineering, Xi'an Jiaotong University, Xi'an, China. ✉e-mail: bcui@stanford.edu

of curved adhesions is driven by membrane curvatures rather than mechanical forces, which allows cells to adhere to soft ECM fibres in 3D environments.

## Results

### Membrane curvatures induce the accumulation of integrin β5

We used a recently developed nanostructure platform to induce well-defined membrane curvatures in live cells[23]. We first engineered SiO₂ nanobars, which are vertically aligned with 200 nm width, 2 µm length, 1.4 µm height and 5 µm spacing (Fig. 1a,b, Extended Data Fig. 1a and Supplementary Table 1). When the plasma membrane wraps around nanobars, the two vertical ends and the horizontal top of nanobar surfaces induce well-defined local membrane curvatures[24] (Fig. 1c). In 2D images focused on the middle height of nanobars, owing to the 3D-to-2D projection effect, the curvature effect is primarily observed at the ends of nanobars, with minimal contribution from the top surface. The flat side walls of the nanobars serve as an internal control (Fig. 1b).

In mammals, integrins consist of 18 α-subunits and 8 β-subunits, forming 24 distinct αβ heterodimers that bind to various ECM ligand proteins[25]. To investigate different integrin isoforms, we coated nanobar substrates with their respective ECM ligands (Fig. 1d). Specifically, hydrolysed collagen (gelatin), laminin, fibronectin and vitronectin were used to probe β1-containing integrins, fibronectin and vitronectin to probe β3-, β5-, β6- and β8-containing integrins, and laminin to probe β4-containing integrin. We did not investigate the leukocyte-specific integrins β2 and β7, which primarily mediate cell–cell adhesions. To create uniform ECM coatings, we applied poly-ʟ-lysine (PLL) and crosslinked it to specific ECM proteins using glutaraldehyde, as demonstrated by fluorescent gelatin (Fig. 1e). We used immunofluorescence to probe endogenous integrin β1 (ITGβ1) and integrin β5 (ITGβ5), both of which are highly expressed in U2OS cells[26] (Supplementary Table 2), and carboxy-terminal green fluorescent protein (GFP) tagging to probe integrins β3, β4, β5, β6 and β8.

On flat areas, anti-ITGβ1, β3–GFP, β5–GFP and β6–GFP formed focal adhesions on substrates coated with their respective ECM ligands (Extended Data Fig. 1b). β4–GFP formed hemidesmosomes on laminin, whereas β8–GFP appeared diffusive owing to the lack of talin-binding motifs. On nanobar areas, anti-ITGβ1 localized to focal adhesions between nanobars but showed minimal signals on nanobars, which induce membrane wrapping as visualized by the membrane marker RFP–CaaX (Fig. 1f). Interestingly, in addition to being present in focal adhesions between nanobars, anti-ITGβ5 showed selective and strong accumulation at some nanobar ends, which are locations of high membrane curvature. By contrast, the co-expressed RFP–CaaX was relatively evenly distributed on the same nanobars (Fig. 1g).

On nanobars, β3–GFP, β6–GFP and β8–GFP on fibronectin, as well as β4–GFP on laminin, showed an even distribution along the

nanobar length, similar to RFP–CaaX. However, ITGβ5–GFP on both vitronectin and fibronectin showed selective accumulation at nanobar ends (Fig. 1h). Additionally, anti-ITGβ1 on fibronectin, vitronectin and laminin, as well as β3–GFP, β6–GFP and β8-GFP on vitronectin, showed no preference for nanobar ends (Extended Data Fig. 2a,b). We quantified the curvature preference by measuring the nanobar end/side ratio of integrin β-subunits, normalized by the end/side ratio of RFP–CaaX at the same nanobars. The membrane normalization step is crucial to distinguish the curvature effect from occasional uneven membrane wrapping. The quantifications confirmed that ITGβ5 shows a strong curvature preference, with an average end/side ratio of ~1.9 on both vitronectin and fibronectin (Fig. 1i). The end/side ratios of other β-subunits were ~1.0 on their respective ligands, which indicated that these subunits have no curvature preference.

ITGβ5 interacts with the αv subunit to form an αvβ5 heterodimer for ligand binding. As fibronectin is a weak ligand for αvβ5 but a potent ligand for integrin α5β1, which is also highly expressed in U2OS cells[26], we used vitronectin coating for the subsequent ITGβ5 studies unless noted otherwise. Anti-αvβ5 staining showed preferential accumulation of αvβ5 heterodimers at nanobar ends (Extended Data Fig. 2c). In addition to U2OS cells, preferential accumulation of ITGβ5 at nanobar ends was observed in many human cell lines, including HT1080, A549, U-251, MCF7, HeLa and human mesenchymal stem cells (hMSCs), and in mouse embryonic fibroblast cells (Extended Data Fig. 2d,e). This result indicates that the curvature preference of ITGβ5 is not limited to a specific cell type.

To determine the range of curvatures that induce ITGβ5 accumulation, we engineered gradient nanobars with end-curvature diameters ranging from 100 nm to 5 µm (Extended Data Fig. 1c). Compared to the membrane maker GFP–CaaX, anti-ITGβ5 exhibited a clear preference for the ends of thin nanobars, but the curvature preference gradually diminished as the nanobars became thicker (Fig. 1j). Quantification of the ITGβ5 end/side ratios showed that ITGβ5 preferred sharp curvatures with a curvature diameter of ≤3 µm (Fig. 1k).

### Curved adhesions recruit talin-1 and bear mechanical forces

To determine whether ITGβ5 at curved locations participate in cell–matrix adhesions, we used SiO₂ nanopillar arrays that induce more curvatures per cell than nanobars (Fig. 2a,b and Extended Data Fig. 1d). ITGβ5–GFP preferentially accumulated at some membrane-wrapped nanopillars and in focal adhesions formed between nanopillars on vitronectin-coated substrates. However, it appeared diffusive on the plasma membrane when the nanopillar substrates were coated with a non-ligand gelatin (Fig. 2c and Extended Data Fig. 3a). Quantification of ITGβ5–GFP, normalized to RFP–CaaX, confirmed that ITGβ5–GFP preferentially accumulated on vitronectin-coated but not gelatin-coated nanopillars (Fig. 2d). When extracellular Ca²⁺, which is required for the

**Fig. 1 | Positive membrane curvature induces selective accumulation of integrin β5. a**, SEM image of nanobars viewed at a 45° angle. Scale bar, 5 µm. **b**, Single nanobar viewed at a 45° angle (top) and from the top (bottom). Scale bar, 1 µm. **c**, SEM image showing cell membrane deformation on nanobars. Scale bar, 5 µm (full size) or 1 µm (inset). **d**, Chart showing the integrin β-subunits probed in this study, with their respective α-subunits and ECM ligands. **e**, Left: schematic of an ECM ligand-coated surface. Right: bright-field and fluorescence images of nanobars coated with Atto647–gelatin. Scale bar, 5 µm. GA, glutaraldehyde. **f**, Anti-ITGβ1 on gelatin-coated nanobar substrate does not show accumulation at nanobars visualized by a membrane marker RFP–CaaX transiently expressed in some cells. Scale bar, 10 µm (full size) or 5 µm (insets). Arrowheads indicate nanobar ends. **g**, Anti-ITGβ5 on vitronectin-coated nanobar substrate shows preferential accumulation at the nanobar ends (arrowheads), whereas RFP–CaaX is relatively evenly distributed along the same nanobars. Scale bar, 10 µm (full size) or 5 µm (insets). **h**, Fluorescence images of GFP-tagged β3, β4, β5, β6 and β8 integrins co-expressed with RFP–CaaX on nanobar substrates with their respective ECM protein coatings. Only ITGβ5–GFP shows

preferential accumulation at nanobar ends (arrowheads). Scale bar, 10 µm (full size) or 5 µm (insets). More images are included in Extended Data Fig. 2a. **i**, Quantifications of curvature preferences of integrin β-subunits by measuring their nanobar end/side ratios, normalized by the end/side ratios of RFP–CaaX at the same nanobars. Each data point represents the mean value from a single cell having between 26 and 158 nanobars (see source data for Fig. 1). *n* = 12 cells, pooled from 2 independent experiments per condition. **j**, Probing the curvature range that induces ITGβ5 accumulation using gradient nanobar arrays. First row: bright-field (BF) image of gradient nanobars. Second row: fluorescence image of gradient nanobars coated with Cy3–vitronectin (Vn). Third and fourth rows: anti-ITGβ5 in cells expressing GFP–CaaX on vitronectin-coated gradient nanobars. Arrowheads indicate nanobar ends. Scale bar, 10 µm. **k**, Quantification of the end/side ratio of ITGβ5/CaaX on gradient nanobars. *n* = 8 images for each condition, from 2 independent experiments. *P* values calculated using one-way ANOVA with Bonferroni's multiple comparison. Data are the mean ± s.d. Source numerical data are available in the source data.

ligand binding of integrins[27], was sequestered with ethylenediaminetetraacetic acid (EDTA), ITGβ5–GFP accumulation on vitronectin-coated nanopillars was abolished (Extended Data Fig. 3b,c). As the αvβ5 accumulation at curved membranes requires ECM binding, we term these structures curved adhesions.

We performed live-cell imaging to determine the dynamics of curved adhesions. ITGβ5–GFP, compared to the membrane marker CellMask, selectively accumulated on ~30% of nanopillars (Fig. 2e). The majority of ITGβ5–GFP-marked curved adhesions (24 out of 28 nanopillars in the image) persisted for >80 min (Fig. 2e,f and

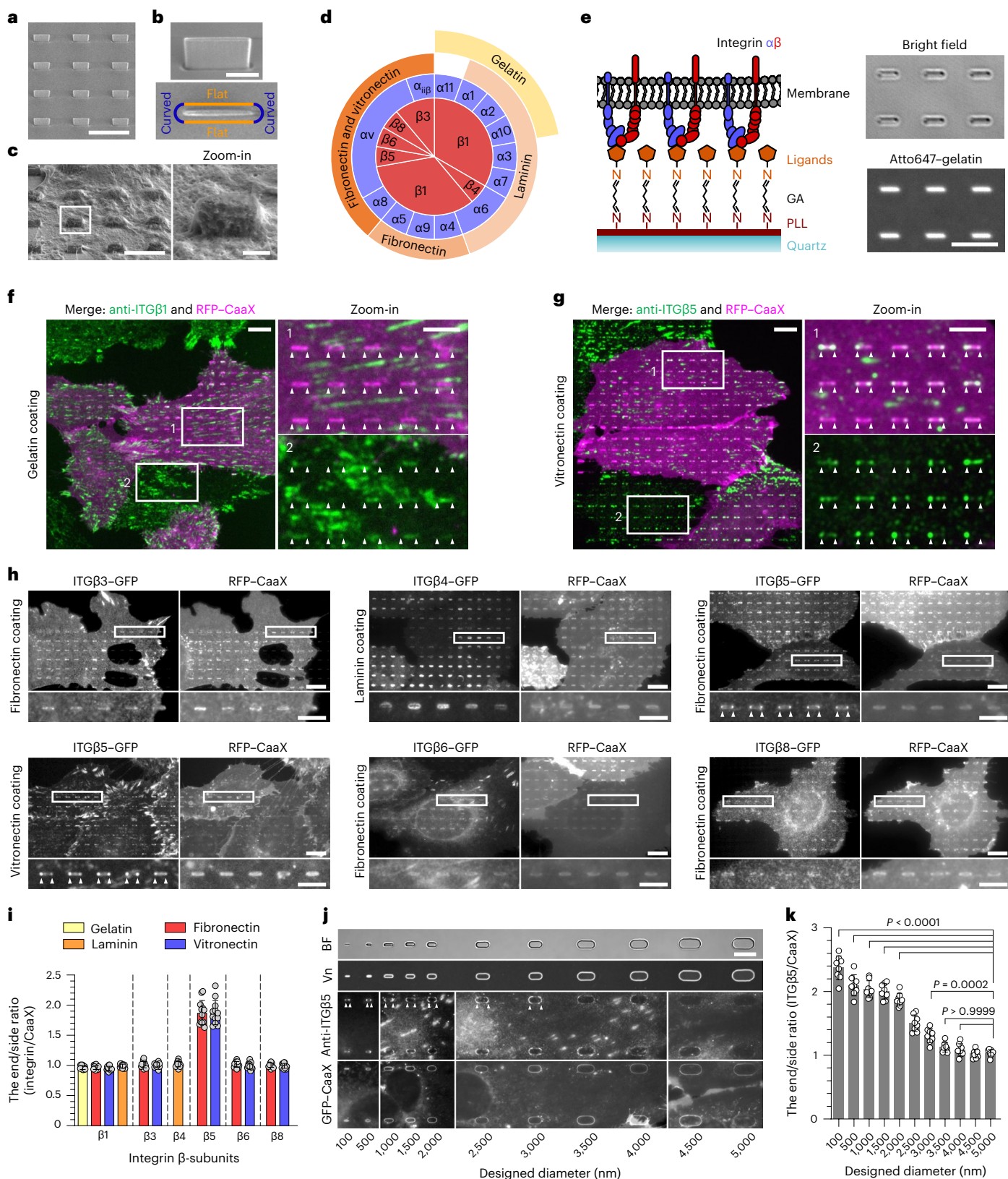

Supplementary Video 1). Nevertheless, slow and gradual assembly (two nanopillars, example red trace in Fig. 2g) and disassembly (three nanopillars, example blue trace in Fig. 2g) of curved adhesions occurred at a few nanopillars without significant changes in membrane wrapping (Fig. 2g and Extended Data Fig. 3d). This dynamic measurement shows that curved adhesions are stable adhesions.

To explore whether curved adhesions can bear mechanical forces, we first examined whether they could recruit talin-1, a key force transmission component of focal adhesions[4]. Immunofluorescence analyses revealed that talin-1 selectively assembled at the nanopillars where ITGβ5 accumulated (Fig. 2h). In the same image, talin-1 also colocalized with ITGβ5 in focal adhesions as expected. By contrast, ITGβ1 did not accumulate at vitronectin-coated nanopillars where talin-1 clearly accumulated (Extended Data Fig. 3e). Additionally, with gelatin coating, talin-1, ITGβ5 and ITGβ1 did not accumulate at the nanopillars even though talin-1 and ITGβ1 colocalized in focal adhesions (Extended Data Fig. 3f,g). These data demonstrate that curved adhesions involve talin-1 and are specific to ITGβ5.

To determine whether the talin-1 molecules in curved adhesions are under tension, we used two genetically encoded and Förster resonance energy transfer (FRET)-based biosensors, in which a tension sensor module (TSM) is inserted between the head domains and the rod domains of talin-1 (refs. 28,29) (Fig. 2i). When talin-1 is under mechanical forces, the TSM is stretched, resulting in a reduced FRET efficiency. As negative controls, the TSM modules were linked to either the amino terminus or the carboxyl terminus of talin-1, and thus the FRET efficiency was not affected by the forces exerted on talin-1. Specifically, the sensor FL-TSM is sensitive to low forces with a maximal sensitivity 3–5 pN, whereas the sensor HPst-TSM is sensitive to high forces with a maximal sensitivity 9–11 pN. The talin tension sensors were incorporated in both curved adhesions at nanopillars and focal adhesions (YPet fluorescence in Fig. 2j). We quantified the normalized FRET ratio using ratiometric imaging (Methods). The low-force FL-TSM sensor exhibited similar FRET ratios at curved adhesions and focal adhesions, which were significantly lower than its controls (Fig. 2i,j and Extended Data Fig. 3h), which indicated that the FL-TSM sensor is stretched in both adhesion architectures. When the high-force HPst-TSM sensor was used, focal adhesions showed significantly lower FRET ratios than curved adhesions, which indicated that there was a greater extent of TSM stretching in focal adhesions. Together, these results indicate that curved adhesions bear mechanical forces, and the tension on talin-1 is lower in curved adhesions than in focal adhesions.

Besides bearing forces, curved adhesions promote early-stage cell spreading, which is a characteristic of functional cell adhesions[30]. Immunofluorescence analyses showed strong ITGβ5 accumulation at vitronectin-coated nanopillars 30 min after cell plating, which is before the formation of large focal adhesions (Fig. 2k and Extended Data Fig. 4a). Under this condition, cells at nanopillar areas were visibly larger than those at flat areas in the same images (Fig. 2l). Similar results were observed for cells cultured on fibronectin-coated substrates, but not on substrates coated with gelatin, PLL or bovine serum albumin (BSA) (Fig. 2m and Extended Data Fig. 4b). ITGβ5 knockdown cells showed no nanopillar-assisted early-stage cell spreading on either vitronectin-coated or fibronectin-coated substrates (Extended Data Fig. 4a,c; knockdown efficiency verified in Fig. 6b).

## Curved adhesions involve a subset of focal adhesion proteins

To investigate whether curved adhesions involve other adhesion proteins alongside talin-1, we examined the spatial correlations between ITGβ5 and well-known adhesion proteins, including paxillin, vinculin, phospho-focal adhesion kinase (Tyr397, pFAK) and zyxin. ITGβ5 colocalized with all of these proteins in focal adhesions (Extended Data Fig. 4d–g). However, vinculin and pFAK were absent from ITGβ5-marked curved adhesions at nanopillars (Fig. 3a and Extended Data Fig. 4h), even though their colocalization with ITGβ5 in focal adhesions formed between nanopillars in the same images. By contrast, paxillin and zyxin colocalized with ITGβ5, similar to talin-1, in both curved adhesions and focal adhesions (Extended Data Fig. 4i,j). The involvement of talin-1 and zyxin, but not vinculin, in curved adhesions was further confirmed by their preferential accumulation at nanobar ends (Extended Data Fig. 4k). Quantifications of the Spearman's correlation coefficients between ITGβ5 and adhesion proteins at nanopillars suggested that curved adhesions contain a distinct subset of adhesion proteins, including talin-1, paxillin and zyxin, but not vinculin or pFAK (Fig. 3b). Additionally, curved adhesions were not linked to thick stress fibres, but were connected to local F-actin assemblies at nanopillars[31] (data and discussions in Extended Data Fig. 5).

Next, we explored whether curved adhesions are related to clathrin-containing adhesions such as flat clathrin lattices and reticular adhesions[17]. Clathrin-containing structures have been reported to recruit β5 as well as β1 and β3 integrins[17–20,32]. We observed that both ITGβ5 and AP2, the clathrin adaptor, accumulated at vitronectin-coated nanopillars (Fig. 3c). However, closer examination showed that their accumulations were not correlated (Fig. 3c and quantification in Fig. 3b). For instance, nanopillars with high β5 intensities often had low AP2 intensities and vice versa. Even when both ITGβ5 and AP2 accumulated at the same nanopillars, expansion microscopy imaging of the cell–nanopillar interface[33] showed that they were not spatially correlated along nanopillars (Extended Data Fig. 6a). Similarly, ITGβ5–GFP accumulation at nanopillars did not correlate with the anti-clathrin heavy chain (anti-CHC) or another clathrin adaptor protein, EPS15–RFP (Extended Data Fig. 6b and quantifications in Fig. 3b).

**Fig. 2 | Curved adhesions recruit talin-1 and bear low mechanical forces. a**, An SEM image of nanopillars. Scale bar, 1 µm. **b**, Left: zoom-in on a single nanopillar in **a**. Scale bar, 1 µm. Right: Schematic of integrin adhesion at a nanopillar. **c**, Fluorescence images showing that vitronectin-coated but not gelatin-coated nanopillars induce ITGβ5–GFP accumulation relative to RFP–CaaX. Ratiometric images are shown in the Parula colour scale. Scale bar, 10 µm. **d**, Quantifications of the normalized ITGβ5/CaaX ratio at individual nanopillars and their flat surrounding regions. Left to right, $n$ = 926, 926, 863 and 863 pillars, from 3 independent cells. Medians (lines) and quartiles (dotted lines) are shown. **e**, Live-cell imaging of ITGβ5–GFP with CellMask membrane marker on vitronectin-coated nanopillars at 15 s per frame for 80 min. Ratiometric images at 0 min and 80 min are shown. Scale bar, 10 µm. **f**, Kymograph of the rectangular box in **e**. Scale bar, 1 µm. **g**, Example trajectories of ITGβ5 nanopillar accumulations showing that most curved adhesions persist, with a few that slowly assemble (pillar 1) or disassemble (pillar 2). **h**, On vitronectin-coated substrates, endogenous ITGβ5 and talin-1 colocalize in both curved adhesions at nanopillars (arrows) and focal adhesions between nanopillars on flat areas (arrowheads). Scale bar, 10 µm (full size) or 5 µm (insets). **i**, Top: schematic of talin-1 tension sensors. mCh, mCherry. Bottom: quantification of the normalized

average FRET ratio (in arbitrary units (a.u.)) at focal adhesions and curved adhesions. Left to right, $n$ = 13, 12, 10, 10, 15 and 11 cells, from 2 independent experiments. **j**, Fluorescence (YPet) and ratiometric FRET images (in the Parula colour scale) of the low-force FL-TSM sensor and the high-force HPst-TSM sensor. In the zoom-in window of the HPst-TSM sensor, curved adhesions exhibit higher FRET values, and thus lower tensions, than focal adhesions. Scale bar, 10 µm (full size) or 5 µm (insets). **k**, ITGβ5 accumulates at vitronectin-coated nanopillars within 30 min after seeding. Scale bar, 10 µm. **l**, Early-stage cell spreading (30 min after plating) on vitronectin-coated nanopillar and flat areas in the same image. Scale bar, 50 µm. **m**, Quantification of the cell spreading area. $n$ are as follows (left to right): $n$ = 69, 68, 79, 53, 45 and 48 (vitronectin) and $n$ = 76, 64, 57, 55, 45 and 42 (fibronectin) cells, from 3 independent experiments; $n$ = 38 and 44 (gelatin), $n$ = 35 and 38 (PLL) and $n$ = 42 and 34 (BSA) cells, from 2 independent experiments. Data are the mean ± s.d. (**i**,**m**). $P$ values calculated using Kruskal–Wallis test with Dunn's multiple comparison (**d**,**m** (vitronectin, fibronectin)), one-way ANOVA with Tukey's multiple comparison (**i**), Mann–Whitney test (**m**, gelatin) or two-tailed $t$-test (**m**, PLL, BSA). Source numerical data are available in the source data.

Furthermore, the short hairpin RNA (shRNA)-mediated knockdown of key proteins involved in clathrin-containing adhesions, such as CHC, AP2 μ-subunit (AP2-μ), EPS15 and EPS15R (EPS15/R), or intersectin 1 and intersectin 2 (ITSN1/2), did not affect the curvature preference of ITGβ5–GFP (Extended Data Fig. 6c,d and quantifications in Fig. 3d). Therefore, curved adhesions are molecularly distinct from clathrin-containing adhesions. This result is further supported by the

presence of talin-1, paxillin, zyxin and F-actin in curved adhesions, whereas clathrin-containing adhesions are devoid of talin-1 and other mechanotransduction components[17].

To determine which domain of ITGβ5 is responsible for its curvature preference, we constructed two chimeric proteins, Exβ5/Inβ3 and Exβ3/Inβ5, by swapping the extracellular domains of ITGβ5 and ITGβ3 (Fig. 3e). Exβ5/Inβ3–GFP formed strong focal adhesions on

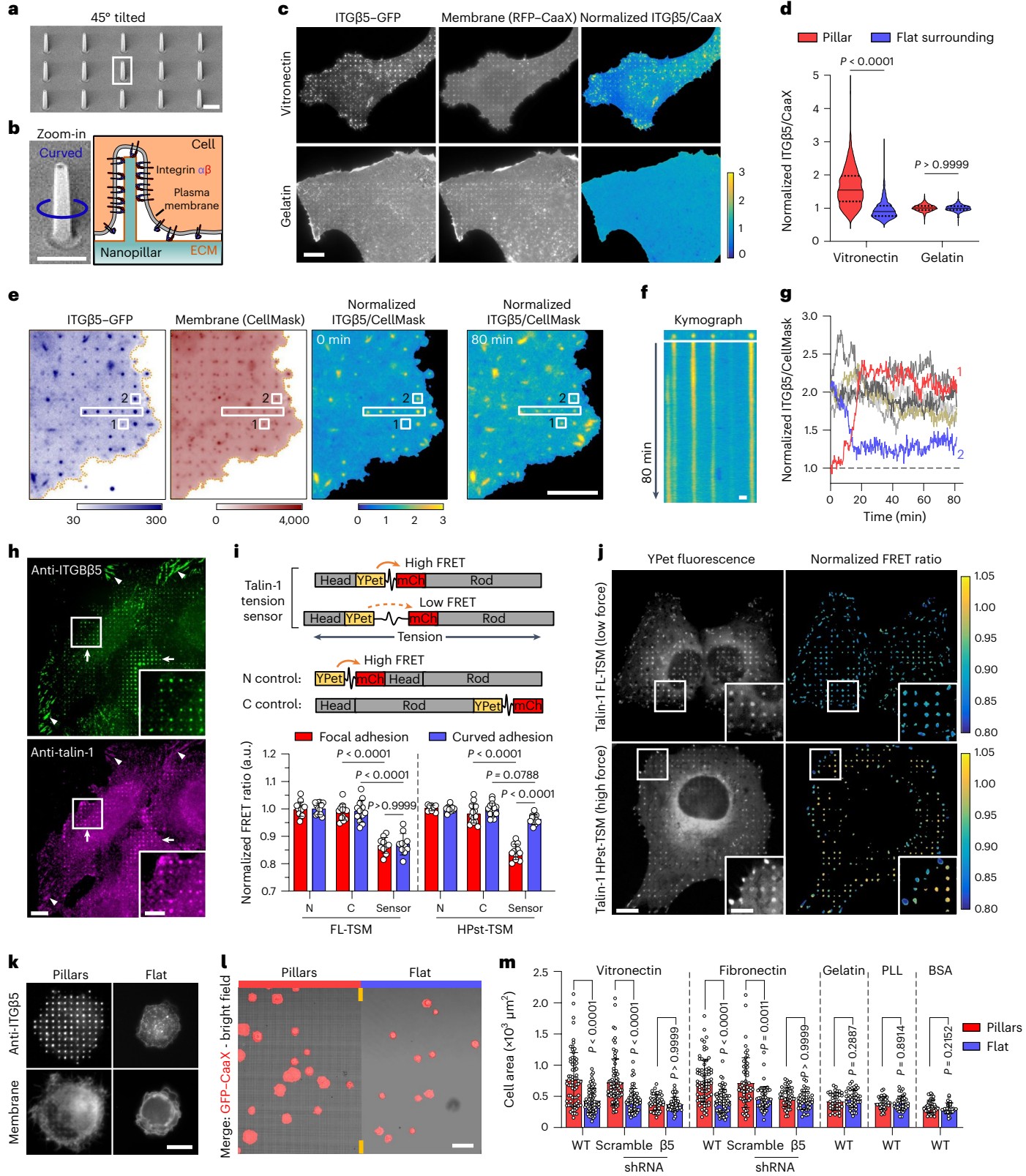

vitronectin-coated substrates but did not exhibit a curvature preference at nanobar ends (Fig. 3f and quantification in Fig. 3i). By contrast, Exβ3/Inβ5–GFP, although not well expressed, showed clear curvature preference on fibronectin-coated nanobars (Fig. 3f and quantification in Fig. 3i). These results indicate that the transmembrane (TM) domain and cytoplasmic domain of ITGβ5, rather than its extracellular domains, are crucial for its curvature preference.

Next, we sequentially truncated the ITGβ5 cytoplasmic domain from the C terminus, resulting in β5(1–779), β5(1–769), β5(1–759) and β5(1–749) (Fig. 3g). With the exception of β5(1–749), all truncated ITGβ5 variants showed curvature preference towards nanobar ends (Fig. 3h and quantification in Fig. 3i). Notably, β5(1–769) and β5(1–759), for which the NPLY talin-binding motif[3] is deleted, could not participate in focal adhesions but still showed clear curvature preference. Therefore, the curvature preference of ITGβ5 does not rely on talin binding but instead requires its intracellular juxtamembrane region.

## Curved adhesions require a curvature-sensing protein FCHo2

When we were investigating the relationship between endocytic proteins and curved adhesion, we found that the shRNA knockdown of FCHo1 and FCHo2, two homologous early-stage endocytic proteins, significantly reduced ITGβ5 accumulation at nanobar ends (Extended Data Fig. 7a). FCHo1 and FCHo2 are intrinsic curvature-sensing proteins that possess an N-terminal F-BAR domain[34]. FCHo1 and FCHo2 are not present in the adhesome of either focal adhesions[35] or clathrin-containing adhesions[21].

RFP–FCHo2 selectively colocalized with ITGβ5–GFP at nanopillars (arrows) but did not colocalize with ITGβ5 in focal adhesions (arrowheads) or clathrin-containing adhesions (thin arrows) in the same cells (Fig. 4a). At nanopillars, the intensities of FCHo2 and ITGβ5 were strongly correlated, with a Spearman's correlation coefficient of ~0.6 (Fig. 4b). Live-cell imaging showed that FCHo2 accumulation was strong and stable in curved adhesions marked by ITGβ5 (Fig. 4c and Supplementary Video 2). Some nanopillars without ITGβ5 also showed weak FCHo2 accumulation, exhibiting frequent and dynamic assembly and disassembly, which is probably associated with endocytosis or other dynamic processes (Fig. 4c,d). The shRNA knockdown of FCHo2, marked by BFP expression, resulted in significantly reduced ITGβ5 accumulation on nanopillars, which is in contrast to non-transfected cells in the same images (Fig. 4e and quantification in Fig. 4g). Interestingly, when ITGβ5 was knocked down through shRNA, FCHo2 accumulation at nanobar ends was also significantly reduced (Extended Data Fig. 7b,c). These results suggest that FCHo2 is a stable and essential component of curved adhesion.

Interestingly, RFP–FCHo1 did not show strong accumulation at nanopillars (Fig. 4f). Even when RFP–FCHo1 accumulated at some nanopillars, it showed little correlation with anti-ITGβ5. Furthermore, the shRNA knockdown of FCHo1 did not affect ITGβ5 accumulation at nanopillars (Extended Data Fig. 7d and quantification in Fig. 4g), which suggested that FCHo1 is not involved in curved adhesions. These results indicate that FCHo2, but not FCHo1, participates in curved adhesions. GFP–CaaX showed that membrane wrapping around nanopillars was not affected by the knockdown of FCHo1 and FCHo2 (Extended Data Fig. 7e).

FCHo2 is composed of a curvature-sensitive F-BAR domain, an intrinsically disordered region (IDR) that is crucial for activating AP2 in clathrin-mediated endocytosis, and a C-terminal μHD domain[36] (Fig. 4h). We constructed six GFP-tagged FCHo2 variants that lack one or two of its three domains and examined their correlation with ITGβ5 at nanopillars. FCHo2_ΔIDR still strongly colocalized with anti-ITGβ5 in curved adhesions, but all other FCHo2 variants had either no correlation or reduced correlation with anti-ITGβ5 (Fig. 4i,j). Overexpression of FCHo2_F-BAR–GFP also induced a significant reduction in anti-ITGβ5 at nanopillars, which is probably due to a dominant-negative effect (Fig. 4i). These data indicate that both the F-BAR domain and the μHD

domain, but not the IDR, are necessary for the participation of FCHo2 in curved adhesions.

Finally, we investigated whether the cytoplasmic region of ITGβ5 interacts with FCHo2. We identified the juxtamembrane region of the ITGβ5 as a crucial region for curvature preference. Therefore, we engineered the fusion protein GFP–β5(715–769), which consists of an extracellular GFP tag, the TM domain and a 27-amino-acid juxtamembrane fragment. We also constructed a negative control, GFP–β5TM, which lacks the crucial juxtamembrane fragment (Fig. 4k). Both GFP–β5(715–769) and GFP–β5TM exhibited plasma membrane localization with some puncta in the perinuclear regions (Fig. 4l). These perinuclear puncta colocalized with a Golgi marker, Golgi–RFP (Extended Data Fig. 8a), which is typical of truncated membrane proteins.

When FCHo2_μHD–RFP was co-expressed with GFP–β5TM, it was diffusive in the cytosol and showed no colocalization with GFP–β5TM (Fig. 4l, top). However, when FCHo2_μHD–RFP was co-expressed with GFP–β5(715–769), it colocalized with GFP–β5(715–769) at perinuclear puncta (Fig. 4l, bottom). Furthermore, GFP–β5(715–769) co-expression caused membrane and perinuclear localization for the other μHD-domain-containing FCHo2 variants (full-length FCHo2, FCHo2_ΔF-BAR and FCHo2_ΔIDR (Extended Data Fig. 8b)), but not for the variants that lack the μHD domain (FCHo2_ΔμHD, FCHo2_F-BAR and FCHo2_IDR (Extended Data Fig. 8c and Fig. 4m)). In control experiments in which GFP–β5TM was co-expressed, full-length FCHo2, like FCHo2_μHD–RFP, did not colocalize with GFP–β5TM (Extended Data Fig. 8d). These results suggest that there is an interaction between the juxtamembrane region of ITGβ5 and the μHD of FCHo2.

To further confirm this interaction, we performed co-immunoprecipitation experiments. When FCHo2_μHD–RFP was co-expressed with GFP–β5(715–769) in HEK293T cells, it immunoprecipitated together with GFP–β5(715–769) (Fig. 4n). In negative controls, no FCHo2_μHD–RFP was co-immunoprecipitated with GFP–β5TM, and no FCHo2_ΔμHD–RFP, a variant lacking the μHD domain, was co-immunoprecipitated with GFP–β5(715–769).

## Curved adhesions abundantly form on 3D ECM protein fibres

Based on their distinct molecular compositions, curved adhesions can be identified by the colocalization of ITGβ5 and FCHo2, whereas focal adhesions can be identified by the colocalization of ITGβ5 and vinculin. On flat and rigid glass substrates (Fig. 5a), numerous focal adhesions were observed (Fig. 5b), whereas minimal curved adhesions were present (Fig. 5c).

The natural ECM is enriched with protein fibres, which may locally deform the cell membrane and induce the formation of curved adhesions. To explore the presence of curved adhesions in physiological environments, we first examined the formation of curved adhesions on a thin layer of fibrous ECM derived from fibroblast cells (Fig. 5d and Extended Data Fig. 9a). Lung fibroblast IMR-90 cells were cultured with ascorbic acid to enhance collagen-based ECM fibre production[37], which resulted in a thin layer (~1 μm thickness) of ECM fibres containing vitronectin, as indicated by the strong anti-vitronectin staining (Fig. 5e), consistent with a previous report[38]. After decellularization, U2OS cells expressing FCHo2–GFP were plated on the cell-derived fibres. Anti-ITGβ5 staining showed two distinct populations: one extensively overlapping with FCHo2–GFP (Fig 5e, arrows) and the other lacking FCHo2–GFP (Fig. 5e, arrowheads). The colocalization of FCHo2 and ITGβ5 indicated the formation of curved adhesions on cell-derived fibres, whereas the population of ITGβ5 patches devoid of FCHo2–GFP probably represents focal adhesions. Notably, curved adhesions consistently aligned with underlying fibres, whereas focal adhesions were often not aligned with and were perpendicular to the fibre direction (Fig. 5e).

Next, we investigated the presence of curved adhesions in 3D ECMs. We made pure collagen fibres containing 10% of AF647-labelled

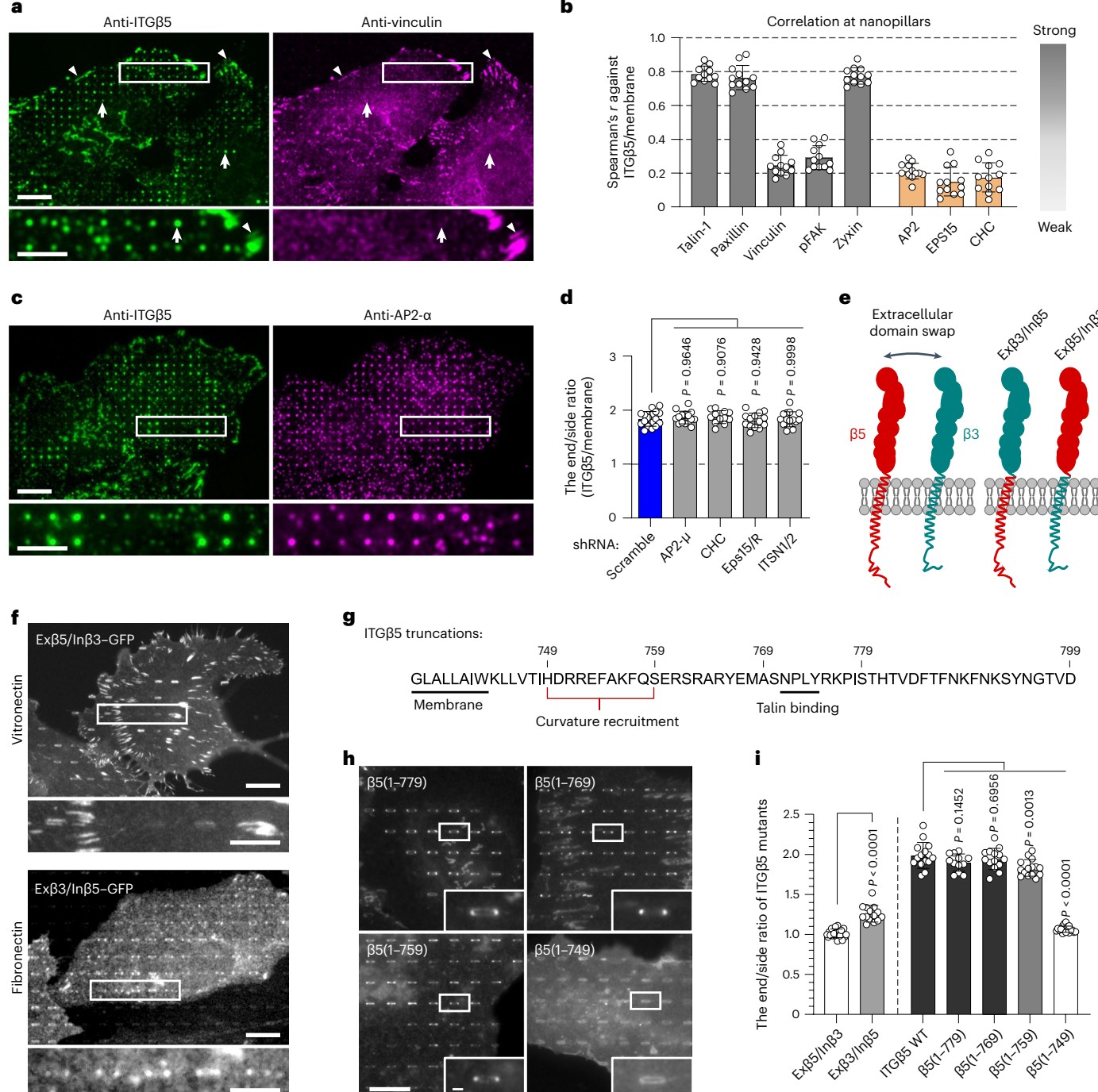

**Fig. 3 | Curved adhesions involve a subset of adhesion proteins and require the juxtamembrane region of ITGβ5 cytoplasmic tail. a**, Vinculin colocalizes with ITGβ5 in focal adhesions (arrowheads) on flat areas but is absent from curved adhesions (arrows) at nanopillars. Scale bar, 10 μm (full size) or 5 μm (insets). **b**, Spearman's correlation coefficients between ITGβ5 and focal adhesion proteins (grey bars) or between ITGβ5 and clathrin-mediated endocytic proteins (orange bars) at nanopillars. Each cell covers between 102 and 769 nanopillars (see source data for Fig. 3). *n* = 12 cells, pooled from 2 independent experiments per condition. **c**, Both ITGβ5 and AP2-α preferentially localize at nanopillars but their intensities are not correlated. Zoom-in images show that nanopillars with high AP2 intensities often have lower ITGβ5 intensities. Scale bar, 10 μm (full size) or 5 μm (insets). **d**, Quantification of ITGβ5 accumulation in curved adhesions (end/side ratio at nanobars) following shRNA knockdown of different endocytic proteins. Left to right, *n* = 17, 15, 12, 14 and 15 cells, from 2 independent experiments. **e**, Illustration of the construction of chimeric

Exβ5/Inβ3 and Exβ3/Inβ5 proteins. **f**, Exβ5/Inβ3 forms prominent focal adhesions but does not accumulate at the ends of nanobars, whereas Exβ3/Inβ5 shows curvature preference for nanobar ends. Scale bar, 10 μm (full size) or 5 μm (insets). **g**, Sequence of the ITGβ5 cytoplasmic domain and the truncation sites. **h**, Fluorescence images of GFP-tagged ITGβ5 truncations β5(1–779), β5(1–769), β5(1–759) and β5(1–749) on vitronectin-coated nanobars. All truncations, except β5(1–749), show curvature preference for nanobar ends. Scale bar, 10 μm (full size) or 1 μm (insets). **i**, Quantification of the curvature preferences of chimeric, wild-type (WT) and truncated ITGβ5 by measuring their nanobar end/side ratio. Left to right, *n* = 19, 14, 15, 12, 14, 15 and 18 cells, from 2 independent experiments. Data are the mean ± s.d. *P* values calculated using one-way ANOVA with Bonferroni's multiple-comparison (**d,i**, WT versus truncations) or two-tailed *t*-test (**i**, Exβ5/Inβ3 versus Exβ3/Inβ5). Source numerical data are available in the source data.

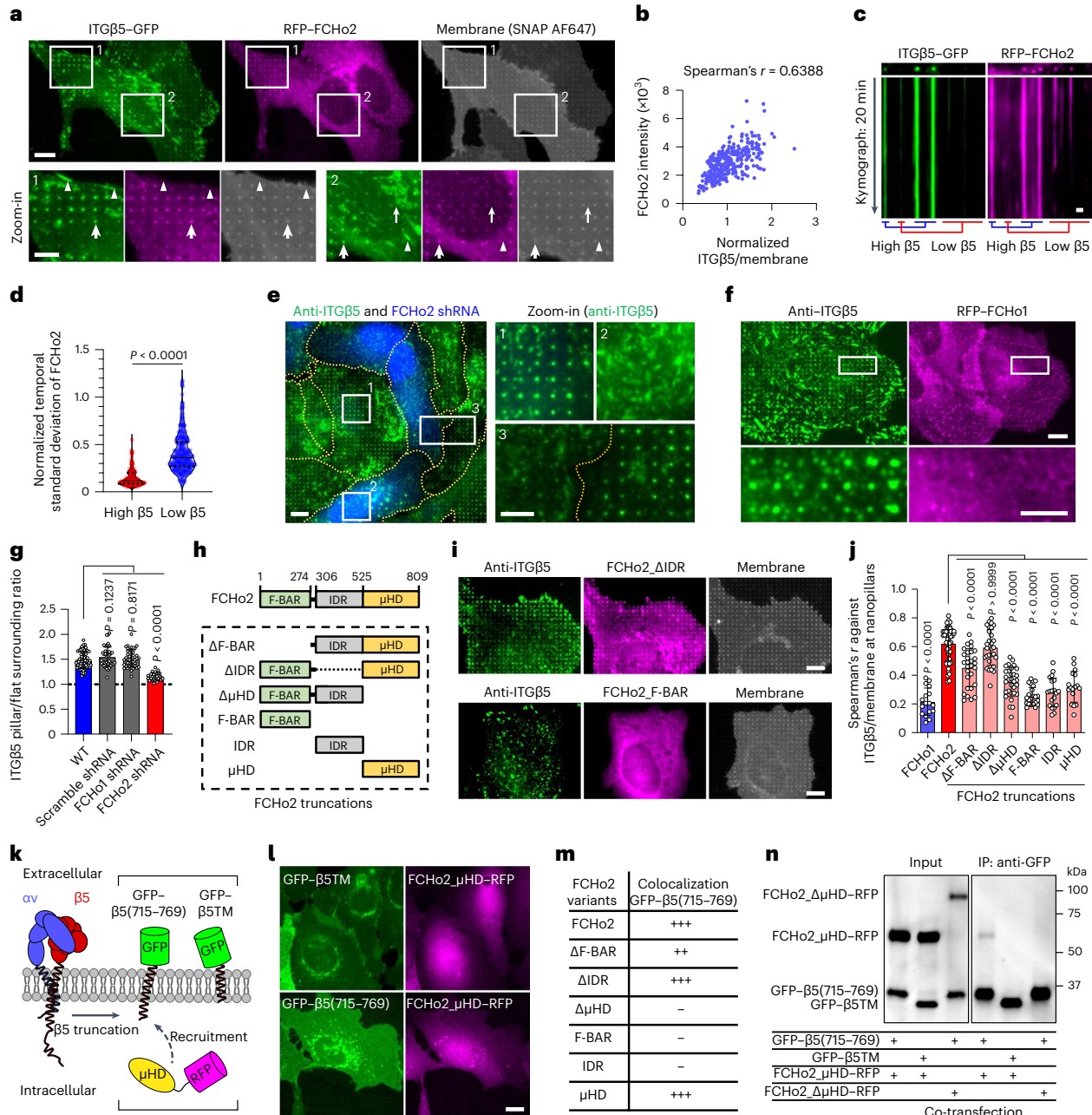

**Fig. 4 | Curved adhesions require a curvature-sensing protein, FCHo2.**
**a**, RFP–FCHo2 positively correlates with ITGβ5–GFP in curved adhesions (arrows) at vitronectin-coated nanopillars. In the same image, FCHo2 is absent in focal adhesions (arrowheads) and clathrin-containing adhesions (thin arrows). Scale bar, 10 μm (full size) or 5 μm (insets). **b**, Scatter plot of RFP–FCHo2 intensity against the ITGβ5/membrane ratio at $n = 353$ nanopillars from the cell shown in **a**. See **g** for the correlation between ITGβ5 and FCHo2 at nanopillars, analysed in multiple cells. **c**, Kymographs of ITGβ5–GFP and RFP–FCHo2 at vitronectin-coated nanopillars imaged at 15 s per frame for 20 min. Scale bar, 1 μm. **d**, The temporal standard deviations of RFP–FCHo2 for 20 min at $n = 104$ high β5 and 313 low β5 nanopillars, from 2 independent cells. For each cell, nanopillars were divided into two groups based on ITGβ5 intensities: high-β5 nanopillars (top 25%) and low-β5 nanopillars (bottom 75%). Standard deviation values are normalized to the initial intensities. Medians (lines) and quartiles (dotted lines) are shown. **e**, FCHo2 knockdown significantly reduces nanopillar-induced ITGβ5 accumulation. BFP expression is a marker of shRNA transfection. Scale bar, 10 μm (full size) or 5 μm (insets). **f**, ITGβ5 does not colocalize with FCHo1 at nanopillars. Scale bar, 10 μm (full size) or 5 μm (insets). **g**, Quantification showing that FCHo2 knockdown significantly reduces nanopillar-induced ITGβ5 accumulation,

but FCHo1 knockdown cannot. Left to right, $n = 68, 43, 44$ and 30 cells, from 2 independent experiments. **h**, Illustration of FCHo2 domain organization and truncations. **i**, Representative images showing that FCHo2_ΔIDR correlates with ITGβ5 in curved adhesions at nanopillars, but FCHo2_F-BAR does not. FCHo2_F-BAR overexpression also reduces ITGβ5 accumulation at nanopillars. Scale bar, 10 μm. **j**, Spearman's correlation coefficients of FCHo1, FCHo2 and truncated FCHo2 with ITGβ5 at vitronectin-coated nanopillars. Left to right, $n = 17, 48, 29, 29, 31, 22, 20$ and 17 cells, from 2 independent experiments. **k**, Illustration of the engineered proteins GFP–β5(715–769) and GFP–β5TM, and the cytosolic protein FCHo2_μHD–RFP. **l**, Top: FCHo2_μHD–RFP is cytosolic and diffusive when co-expressed with GFP–β5TM. Bottom: when co-expressed with GFP–β5(715–769), FCHo2_μHD–RFP colocalizes with GFP–β5(715–769) in the perinuclear Golgi region. Scale bar, 10 μm. **m**, Qualitative analysis of GFP–β5(715–769)-induced membrane re-localization of FCHo2 variants. **n**, Immunoblots of co-immunoprecipitation assay confirms the interaction between the ITGβ5 juxtamembrane region and FCHo2_μHD. Data are the mean ± s.d. (**g**,**j**). $P$ values calculated using Mann–Whitney test (**d**) or one-way ANOVA with Bonferroni's multiple comparison (**g**,**j**). Source numerical data and unprocessed blots are available in the source data.

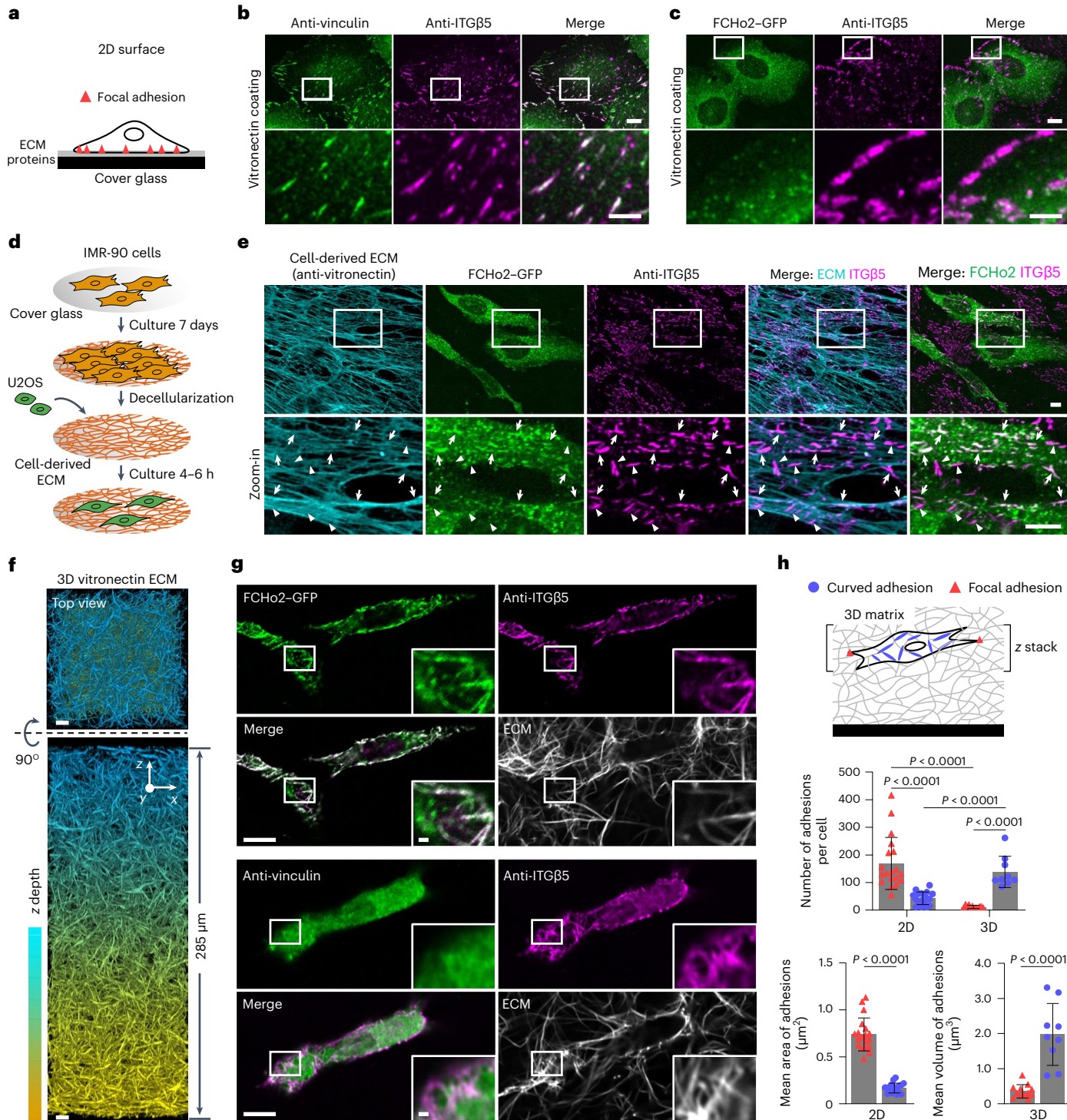

**Fig. 5 | Curved adhesions are prevalent in physiologically relevant environments. a**, Schematic of a cell growing on ECM-coated 2D flat surfaces. **b**, ITGβ5 and vinculin strongly colocalize on vitronectin-coated flat surfaces. Scale bar, 10 μm (full size) or 5 μm (insets). **c**, ITGβ5 and FCHo2 do not colocalize on vitronectin-coated flat surfaces. Scale bar, 10 μm (full size) or 5 μm (insets). **d**, Schematic illustrating the generation of cell-derived ECM fibres. **e**, IMR-90-derived ECM fibres have vitronectin incorporated as shown by anti-vitronectin staining. U2OS cells form both curved adhesions (arrows, colocalization of anti-ITGB5 with FCHo2–GFP) and focal adhesions (arrowheads, anti-ITGB5 devoid of FCHo2–GFP) on these fibres. Scale bar, 10 μm. **f**, Representative 3D images of

a thick layer of matrices made of vitronectin fibres. Scale bar, 10 μm. **g**, Curved adhesions, marked by the colocalizations of ITGβ5 and FCHo2, are abundant on vitronectin fibres in 3D matrices (top), whereas focal adhesions marked by the colocalization of ITGβ5 and vinculin are sparse (bottom). Scale bar, 10 μm (full size) or 1 μm (insets). **h**, Quantification of the number and size of focal adhesions and curved adhesions on 2D flat surfaces and in 3D matrices of vitronectin fibres. Left to right, n = 19, 18, 12 and 9 cells, from 2 independent experiments. Data are the mean ± s.d. P values calculated using two-tailed t-test (**h**, adhesion size) or Kruskal–Wallis test with Dunn's multiple-comparison (**h**, adhesion number). Source numerical data are available in the source data.

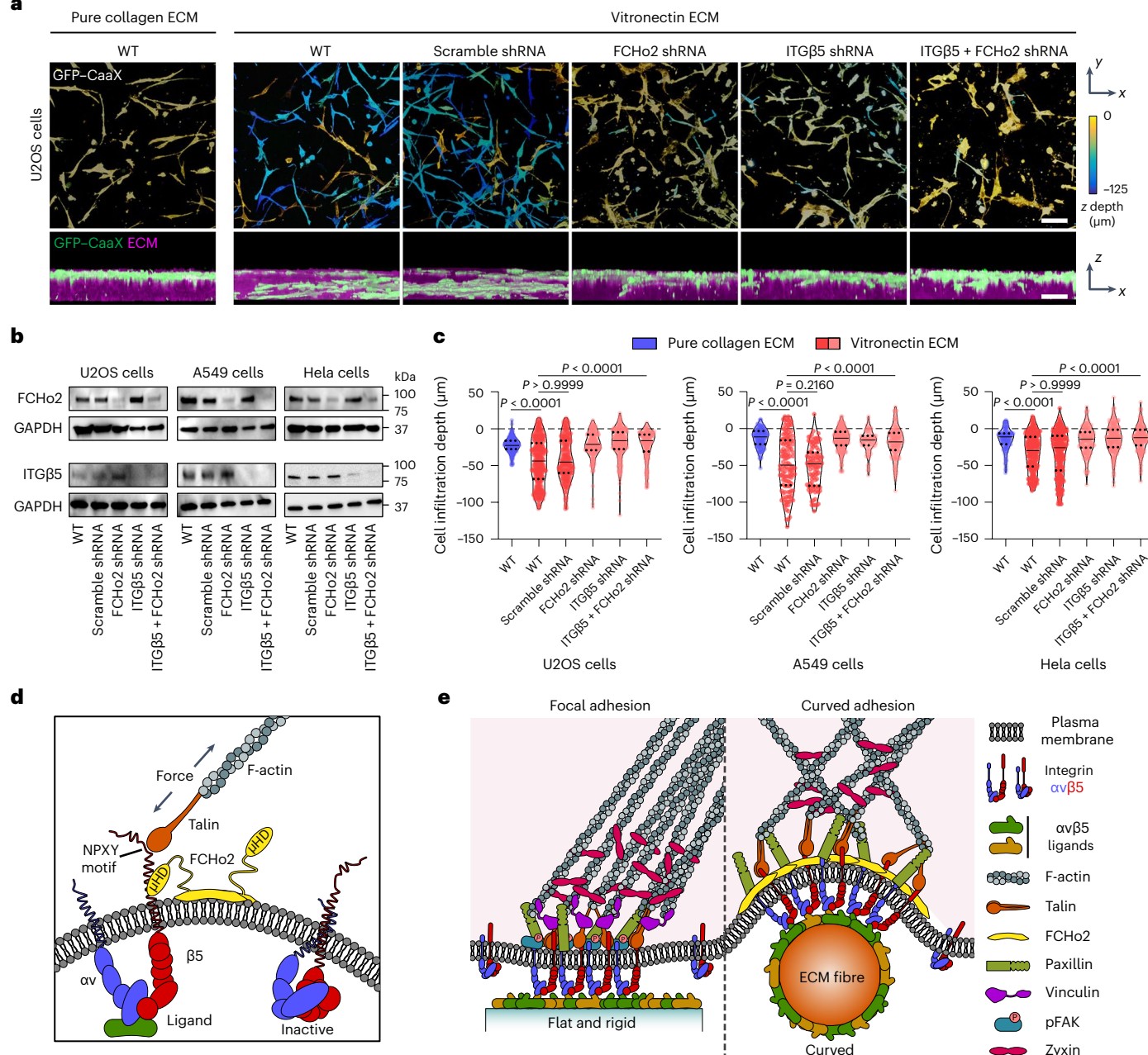

**Fig. 6 | Curved adhesions facilitate cell migration in 3D ECMs.**
**a**, Representative images of U2OS cells expressing GFP–CaaX in 3D matrices after 72 h of culture. Cells are colour-coded according to z depth in x–y projections. Cells are coloured in green and merged with ECM in magenta in x–z projections. WT cells infiltrated into matrices made of vitronectin fibres, but not matrices made of pure collagen fibres. The shRNAs of FCHo2, ITGβ5 or both, but not scramble shRNA, inhibited the cell infiltrations. Scale bar, 100 μm. **b**, Western blots show that shRNAs are able to effectively reduce the expression of FCHo2, ITGβ5 or both in U2OS, A549 and Hela cells. These results support the knockdown efficiencies supplied by the company (Supplementary Table 3). **c**, Quantification

of the cell infiltration depth for U2OS, A549 and Hela cells in 3D pure collagen matrices or 3D vitronectin matrices with the knockdown of FCHo2, ITGβ5 or both. Left to right, $n = 244, 509, 348, 279, 251$ and 211 U2OS cells, $n = 164, 134, 100, 238, 92$ and 177 A549 cells and $n = 142, 236, 277, 189, 241$ and 204 Hela cells, from 2 independent experiments. Medians (lines) and quartiles (dotted lines) are shown. P values calculated using Kruskal–Wallis test with Dunn's multiple comparison. **d**, Illustration of ITGβ5 interacting with FCHo2 in curved adhesions. **e**, Schematic comparison of focal adhesion and curved adhesion architectures. Note that the models only depict proteins investigated in this work. Source numerical data and unprocessed blots are available in the source data.

collagen. Collagen is not a ligand for integrin αvβ5. To study curved adhesions, we prepared vitronectin fibres by incubating collagen fibres with multimeric vitronectin, which has a high affinity for collagen binding[39,40]. Colocalizations of AF647–collagen and anti-vitronectin confirmed the incorporation of vitronectin in collagen fibres (Extended Data Fig. 9b). The most observed diameter of these fibres was ~115 nm as measured by 3D super-resolution microscopy (Extended Data

Fig. 9c,d). Importantly, we used a two-step assembly method to prepare 3D matrices of 200–300 μm thickness (Methods, Fig. 5f and Extended Data Fig. 9e). Cells were plated on top of the 3D matrices. After 72 h of culture, these cells extensively infiltrated and fully embedded themselves in the vitronectin fibre matrices, whereas they mostly remained on the surface of pure collagen fibre matrices (Extended Data Fig. 9f and Supplementary Videos 3 and 4). The deformation

of the plasma membrane by the fibres was evident (Extended Data Fig. 9g,h).

In cells embedded in 3D vitronectin fibre matrices, the anti-ITGβ5 signals significantly overlapped with FCHo2–GFP on fibres (Fig. 5g, Extended Data Fig. 10a and Supplementary Video 5), which indicated the presence of curved adhesions. Conversely, anti-ITGβ5 accumulation on ECM fibres rarely overlapped with anti-vinculin (Fig. 5g, Extended Data Fig. 10b and Supplementary Video 6), which indicated the scarcity of focal adhesions in soft 3D matrices, which agrees with some previous reports[11,12,15]. We quantified curved adhesions by the colocalized signals of ITGβ5 and FCHo2, and focal adhesions by the colocalized signals of ITGβ5 and vinculin. This analysis confirmed the dominance of curved adhesions in the soft 3D matrices (Fig. 5h).

### Curved adhesions facilitate cell migration in 3D ECMs

Because U2OS cells migrate into vitronectin fibre matrices but not pure collagen fibre matrices, we proposed that the formation of curved adhesions facilitates cell migrations in 3D matrices. To test this hypothesis, we perturbed curved adhesions using shRNA lentiviral vectors against FCHo2, ITGβ5 or both. Cells expressing GFP–CaaX were plated on the top surface of 3D fibrous matrices and allowed to migrate for 72 h (Fig. 6a). Wild-type cells migrated deep into vitronectin fibre matrices. However, the perturbation of curved adhesions largely abolished cell migration into vitronectin fibre matrices. Furthermore, when the matrices were made of pure collagen fibres that do not support curved adhesions, wild-type U2OS cells failed to infiltrate deep into the matrices.

We also examined additional cancer cell lines, including A549 cells and HeLa cells. Neither A549 nor HeLa cells migrated into pure collagen fibres matrices after 72 h. By contrast, both cell lines efficiently infiltrated into vitronectin fibre matrices (Extended Data Fig. 10c). Knockdown of FCHo2, ITGβ5 or both blocked cells from migrating into vitronectin fibre matrices. Western blots confirmed the knockdown in all three cell lines (Fig. 6b). Quantification of cell infiltration depth confirmed that the formation of curved adhesions facilitated cell migration in soft 3D matrices (Fig. 6c).

## Discussion

Curved adhesions are cell–matrix adhesions that selectively form on curved membranes and are mediated by integrin αvβ5. Our findings demonstrated that curved adhesions are molecularly and functionally distinct from focal adhesions and clathrin-containing adhesions. Our results support a model whereby curved adhesions require an interaction between the juxtamembrane region of ITGβ5 and the μHD domain of FCHo2, whereas the F-BAR domain of FCHo2 confers curvature sensitivity (Fig. 6d). Curved adhesions involve a distinct subset of adhesion proteins, including talin-1, paxillin and zyxin, but not vinculin or pFAK (Fig. 6e). The presence of talin-1 in curved adhesions enables linkage to the actin cytoskeleton and the transmission of mechanical forces. Further investigations are needed to elucidate the detailed mechanisms underlying FCHo2–ITGβ5 and talin–ITGβ5 interactions in curved adhesions and the role of FCHo2–ITGβ5 interactions in ITGβ5 activation. Interestingly, curved adhesions are abundant in physiologically relevant soft environments and facilitate cell migration in 3D matrices. The identification of curved adhesion provides new leads for understanding the complex cell–ECM coupling in biological and pathological conditions.

## Online content

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

## Methods

### Ethics statement

We obtained hMSCs (adipose-derived, PCS-500-011) from the American Type Culture Collection (ATCC). The hMSCs were not differentiated in this study. They were used in accordance with the guidelines of the IRB/SCRO Panel, which adheres to federal, state and Stanford human stem cell research policies. Research involving non-pluripotent stem cells, including hMSCs, does not require SCRO review.

### Antibodies and shRNA plasmids

All shRNAs were obtained from Sigma-Aldrich, except for the scramble shRNA, which was acquired from Addgene as a gift from D. Sabatini (Addgene, 1864). The knockdown efficiencies of the shRNAs were validated by Sigma-Aldrich using quantitative PCR. The clone identities of the shRNA vectors and their knockdown efficiencies are provided in Supplementary Table 3. Antibodies and their dilutions used in this study are listed in Supplementary Table 4.

### Nanofabrication

Quartz wafers (Silicon Materials, 04Q 525-25-1F-SO) were spin-coated with 275 nm CSAR 6200 and 100 nm Electra 92 (AllResist) e-beam resists. The desired patterns were exposed to e-beam using a JEOL JBX-6300FS system. A 120-nm-thick layer of chromium mask was deposited on the post-exposure chips using an AJA e-beam evaporator and then immediately lifted off with acetone and isopropanol. The Cr mask-patterned chips were etched anisotropically to create vertical nanostructures by reactive ion etching (Plasma Therm Versaline LL ICP Dielectric Etcher, PT-Ox) with a mixture of $C_4F_8$, $H_2$ and Ar for 3 min. Finally, the chips were immersed in chromium etchant 1020 (Transene) for 30 min to remove the Cr mask.

### Fluorescence labelling of gelatin, vitronectin and collagen

To prepare for gelatin labelling, gelatin (Sigma-Aldrich, G9391) was added into water and autoclaved to make a 2 mg ml$^{-1}$ gelatin solution, which was then diluted twofold in 0.1 M sodium bicarbonate buffer. To prepare for vitronectin labelling, human multimeric vitronectin solution (Molecular Innovations, HVN-U) was adjusted to a concentration of 1 mg ml$^{-1}$ and buffer exchanged to 0.1 M sodium bicarbonate using a Zeba spin desalting column (Thermo Scientific, 89882). For fluorescence labelling, Atto 647 NHS ester (Sigma-Aldrich, 07376) or Cy3 mono NHS ester (Sigma-Aldrich, GEPA13101) dye was dissolved in 0.1 M sodium bicarbonate solutions to a final concentration of 1 µg ml$^{-1}$. The dye solution was mixed with the protein solution in a 1:1 molecular ratio and incubated for 1 h at room temperature (RT). Free dyes were then removed, and the buffer was changed to 1x PBS (Gibco) using Zeba spin desalting columns.

For collagen labelling, we adapted a protocol from a previous study[41]. In brief, collagen type I (Corning, 354236) was first diluted into a final concentration of 4 mg ml$^{-1}$ with ice-cold 20 mM acetic acid. The collagen solution (1 ml) was neutralized with a pre-chilled mixture of 20 µl 1 M NaOH, 200 µl 10× DMEM, 200 µl 10 mM HEPES and 580 µl water. The neutralized collagen was dropped into a pre-chilled 10-cm Petri dish and incubated for 1 h at RT for fibre formation. The collagen fibres were washed with 10 ml PBS 3 times for 10 min each and then washed with 10 ml 0.1 M sodium bicarbonate 3 times for 10 min each to remove non-polymerized collagen. Alexa Fluor 647 (AF647) NHS ester (Thermo Scientific, A37573) was dissolved in 0.1 M sodium bicarbonate to a final concentration of 2 µg ml$^{-1}$ and then added to the collagen fibres. In this way, the dye labels collagen at locations that do not interfere with collagen polymerization. After 30 min of incubation at RT, the free dye was washed away with PBS 5 times for 15 min each. The labelled collagen fibres were then dissolved in 500 µl of 500 mM acetic acid by overnight incubation at 4 °C. Finally, the buffer was changed back to 20 mM acetic acid by dialysis.

### Surface coating with ECM proteins

The SiO$_2$ chips were initially treated with air plasma (Harrick Plasma) for 15 min and subsequently incubated with 0.1 mg ml$^{-1}$ PLL (Sigma-Aldrich, P5899) in PBS at 37 °C for 1 h to coat the surface with PLL. For coating substrates with other ECM proteins, the PLL-coated nanochips were further incubated with 0.5% (v/v) glutaraldehyde (Sigma-Aldrich, G6257) in PBS at RT for 15 min. After washing with PBS, the chips were incubated in PBS with the desired ECM protein, such as 1 mg ml$^{-1}$ unlabelled gelatin, 1 mg ml$^{-1}$ Atto 647–gelatin, 0.25 mg ml$^{-1}$ human plasma fibronectin (Sigma-Aldrich, 341635), 0.25 mg ml$^{-1}$ unlabelled vitronectin (Pepro-Tech, 140-09), 0.25 mg ml$^{-1}$ Cy3–vitronectin, 0.25 mg ml$^{-1}$ laminin 511 (Sigma-Aldrich, CC160) or 1% (w/v) BSA (Sigma-Aldrich, A9418). The reaction proceeded at 37 °C for 1 h, and in the case of fluorescently labelled proteins, in the dark. Before cell seeding, non-fluorescent substrates were treated with 1 mg ml$^{-1}$ sodium borohydride (Sigma-Aldrich, 452882) for 10 min to reduce autofluorescence.

### Preparation of soft 3D ECMs

To prepare pure collagen fibres, unlabelled collagen (Advanced Bio-Matrix, 5007) and AF647-labelled collagen was mixed in a 9:1 (w/w) ratio at a final concentration of 2 mg ml$^{-1}$ in 20 mM acetic acid. For collagen polymerization, 50 µl collagen mixture was neutralized with a mixture of 1 µl 1 M NaOH, 10 µl 10× DMEM, 10 µl 10 mM HEPES and 29 µl water on ice. The neutralized collagen was dropped onto a Lab-Tek II chambered cover glass (Sigma-Aldrich, Z734853) at 15 µl per well, and incubated at RT for 20 min. The samples were then thoroughly washed with 20 mM sodium phosphate buffer (pH 7.4) 5 times. This washing step removes free collagen and floating fibres, resulting in a thin layer of collagen fibres attached on cover glasses. To generate a thick layer of immobilized collagen fibres, the neutralized collagen mixture was added onto the pre-chilled thin fibre layer at 15 µl per well, and incubated at RT for 45 min. The fibres were then gently washed with 20 mM sodium phosphate buffer (pH 7.4) 5 times. It is important to perform the polymerization in two steps to generate 3D fibres. The first thin fibre layer attached to the surface serves as seeds for the polymerization of the second thick layer. Without these seeds, collagen polymerization mostly occurs in solution rather than on the surface. To generate vitronectin fibres, collagen fibres were incubated in 100 µl of 50 µg ml$^{-1}$ multimeric vitronectin (Molecular Innovations, HVN-U) in 20 mM sodium phosphate buffer for 1 h at RT.

### Preparation of cell-derived ECM fibres

The protocol was adapted from previous studies[38,42]. IMR-90 lung fibroblast cells were cultured in complete cell-culture medium supplied with 110 µg ml$^{-1}$ sodium pyruvate and 100 µg ml$^{-1}$ 2-phospho-L-ascorbic acid trisodium salt (Sigma-Aldrich, 49752) for 5–7 days to produce cell-derived ECM. The cultures were incubated for 20 min at 37 °C in calcium-free and magnesium-free PBS supplied with 5 mM EDTA and 2 M urea to remove IMR-90 cells from these fibres without cell lysis. The samples were then gently washed with PBS 10 times before seeding new U2OS cells or being used for characterization such as anti-vitronectin staining.

### Plasmids

The DNA fragment encoding human ITGβ5 was amplified from the pCX–EGFP β5 integrin receptor (a gift from R. Birge, Addgene, 14996). The DNA fragments encoding human integrin β3 and β8 were amplified from the complementary DNA (cDNA) of U2OS cells. The DNA fragment encoding integrin β4 was amplified from pcDNA3.1/Myc-His beta4 (a gift from F. Giancotti, Addgene, 16039). They were inserted into the pEGFP-N1 vector (Clontech) for the expression of β5–GFP, β3–GFP, β8–GFP and β4–GFP. The expression vector of ITGβ6–GFP was a gift from D. Sheppard (Addgene, 13593).

The expression vector of RFP–talin-1 was a gift from M. Davidson (Addgene, 55139). FL-TSM and HP35st-TSM were gifts from C. Grashoff

(Addgene plasmids 101170 and 101251, respectively). The DNA fragments encoding TSMs were inserted between the head and rod domains of talin-1 at amino acids 447 as talin tension sensors. To build control constructs, TSMs were fused to the N terminus of talin-1 with an 18-amino-acid linker or to the C terminus of talin-1 with a 17-amino-acid linker. The linker design was based on a previous study[43]. The resulting DNA fragments were used to replace the EGFP fragment in the pEGFP-N1 vector for the expression of talin tension sensors and controls. The DNA fragments encoding human zyxin were amplified from the cDNA of U2OS cells and then inserted into the pmCherry-N1 vector (Clontech) for the expression of zyxin–RFP.

The DNA fragments encoding GFP–CaaX and RFP–CaaX were generated by fusing the DNA fragment encoding the CaaX motif of K-Ras (GKKKKKKSKTKCVIM) to the 3′ end of the DNA fragments encoding EGFP and FusionRed, respectively. The resulting DNA fragments were used to replace the EGFP fragment in the pEGFP-N1 vector for expression in mammalian cells. The DNA fragment encoding GFP–CaaX was inserted into pLenti pRRL-SV40(puro)_CMV for lentivirus packaging. The expression vectors of LifeAct–RFP were gifts from M. Davidson (Addgene, 54491). The DNA fragment encoding SNAP-tag was amplified from the pSNAP-tag (m) vector (a gift from New England Biolabs and A. Egana, Addgene, 101135), and then inserted into the pDisplay vector (Invitrogen, V66020) for the expression of cell surface SNAP-tag.

The pLKO.1 vector was used for the transcription of shRNA. The puromycin-resistant sequence in the pLKO.1 vector was replaced with a sequence encoding EBFP2 to fluorescently label cells that were transfected or transduced.

The expression vectors of RFP–FCHo1, RFP–FCHo2 and EPS15–RFP were gifts from C. Merrifield (Addgene, 27690, 27686 and 27696, respectively). The DNA fragment of FCHo2 was amplified and cloned into a pEGFP-N1 vector to generate FCHo2–GFP. To generate GFP-tagged or RFP-tagged FCHo2 truncations, DNA fragments encoding FCHo2_F-BAR (amino acids 1–274), FCHo2_ΔF-BAR (amino acids 275–809), FCHo2_ΔIDR (amino acids 1–305 fused with amino acids 520–809), FCHo2_IDR (amino acids 306–526), FCHo2_μHD (amino acids 520–809) and FCHo2_ΔμHD (amino acids 1–525) were amplified and cloned into the pEGFP-N1 or pmCherry-N1 vector. The key oligonucleotides used for truncated and hybrid proteins are provided in Supplementary Table 3.

## Cell culture

U2OS (ATCC, HTB-96), A549 (ATCC, CCL-185), HeLa (ATCC, CCL-2), and mouse embryonic fibroblast (ATCC, CRL-2991) cells were cultured in cell culture medium, Dulbecco's modified Eagle's medium (DMEM) (Gibco) supplied with 10% (v/v) FBS (Sigma-Aldrich) and 1% (v/v) penicillin and streptomycin (Gibco). IMR-90 lung fibroblast cells (ATCC, CCL-186, a gift from S. Dixon) were maintained in cell culture medium supplied with 110 μg ml$^{-1}$ sodium pyruvate (Gibco) and 100 μg ml$^{-1}$ 2-phospho-L-ascorbic acid trisodium salt (Sigma-Aldrich, 49752) to stimulate the production of cell-derived ECM. HEK293T cells (ATCC, CRL-3216) were cultured in cell culture medium supplemented with 110 μg ml$^{-1}$ sodium pyruvate (Gibco) for lentivirus production. HT-1080 (ATCC, CCL-121), U-251MG (Sigma-Aldrich, 09063001) and MCF7 (ATCC, HTB-22) and hMSCs (PCS-500-011) were cultured in Eagle's minimum essential medium (Gibco) supplied with 10% (v/v) FBS and 1% (v/v) penicillin and streptomycin. All cell lines were maintained at 37 °C in a 5% CO$_2$ atmosphere. Before being seeded for experiments, cells were detached by 10 min of incubation in an enzyme-free cell dissociation buffer (Gibco, 13151014) to avoid enzymatic digestion of integrins.

## Transient cell transfection

For U2OS, A549, HT1080, U251MG, MCF7, hMSCs and mouse embryonic fibroblast cells, transient transfections were achieved by electroporation. Cells grown in 35-mm wells were detached, collected and then resuspended in an electroporation buffer containing 100 μl electroporation buffer II (88 mM KH$_2$PO$_4$ and 14 mM NaHCO$_3$, pH 7.4), 2 μl electroporation buffer I (360 mM ATP and 600 mM MgCl$_2$) and 0.5–2 μg plasmid. The cells were electroporated in 0.2-cm gap electroporation cuvettes by Amaxa Nucleofector II (Lonza) using protocols pre-installed by the manufacturer. To mark the Golgi apparatus, U2OS cells were transfected with N-acetylgalactosaminyltransferase–RFP using CellLight Golgi–RFP, BacMam 2.0 virus (Invitrogen, C10593). For HeLa cells, transient transfection was achieved using Lipofectamine 2000. HeLa cells in 35-mm wells were incubated with 2 ml Opti-MEM-I medium (Gibco, 31985062) supplied with 6 μl Lipofectamine 2000 (Invitrogen) and 1–2 μg plasmid for 2 h. To investigate GFP-tagged integrin β-isoforms, transfected cells were cultured for 3 days before being seeded on ECM-coated substrates. The cells were then cultured for 6 h before imaging.

## Lentiviral transduction and stable cell line generation

For lentivirus packaging, HEK293T cells in 35-mm wells were transfected with 1.5 μg third-generation lentivirus package vector, 0.8 μg psPAX2 (a gift from D. Trono, Addgene, 12260) and 0.7 μg pMD2.G (a gift from D. Trono, Addgene, 12259), mixed with 9 μl Lipofectamine 2000 in 2 mL Opti-MEM-I medium. The transfection medium was replaced with 2 ml of HEK293T culture medium 6 h after transfection. The virus-containing supernatants were collected 24 h after. The cell debris was removed by spin and filtration through 0.45-μm PVDF syringe filter units (Millipore). Lentiviral infections were used to transduce cells with shRNAs. Cells were cultured for 72 h before the next experiments. Stable cell lines expressing GFP–CaaX were generated by lentivirus transduction of wild-type cells followed by antibiotic selection with 2.5 μg ml$^{-1}$ puromycin. The antibiotic pressure was released 3 days before experiments.

## Immunofluorescence labelling

For the immunolabelling of ITGβ1 and αvβ5 with primary antibodies targeting extracellular domains, cells were first snap-chilled in ice-cold 1× HBSS (Gibco, 24020117) buffered with 10 mM HEPES (Gibco, 15630106). The samples were then incubated with β1 or αvβ5 primary antibody for 30 min at 4 °C, followed by 3 washes of 5 min each and fixation in 4% ice-cold paraformaldehyde (PFA) in PBS for 15 min at RT. If talin co-staining was required afterward, the samples were fixed for 10 min with 4% PFA in PHEM buffer (PIPES 60 mM, HEPES 25 mM, EGTA 10 mM, MgCl$_2$ 2 mM, pH 6.9) instead of 4% PFA in PBS. After fixation, samples were permeabilized with 0.1% Triton-X (Sigma-Aldrich, T8787) in PBS for 15 min at RT and then blocked with a blocking buffer (5% BSA in PBS) for 1 h at RT. For the immunolabelling of other proteins, cells were fixed, permeabilized and blocked before being incubated with primary antibodies in the blocking buffer for 1 h at RT.

The primary antibody-labelled samples were then washed with the blocking buffer 3 times for 15 min each and incubated in fluorescently labelled secondary antibodies in the blocking buffer for 1 h at RT. The samples were washed with the blocking buffer 3 times for 15 min each and then with PBS 3 times before imaging. For the immunostaining of cells embedded in 3D fibres, the time durations for antibody incubation and washing were doubled.

## SNAP-tag labelling

To preserve the green and red channels for protein labelling in three-channel imaging, we used transiently transfected extracellular SNAP-tag to label the cell membrane with AF647. In brief, cells were transfected with SNAP-pDisplay. The cells were incubated in CO$_2$-balanced pre-warmed medium supplied with 1 μM O$^6$-benzylguanine-coupled AF647 (NEB, S9136S) for 15 min at 37 °C in a 5% CO$_2$ atmosphere, and subsequently washed 5 times with cell culture medium before fixation.

## Immunoblotting

For immunoblotting, cells were rinsed with ice-cold 1× PBS and lysed for 1 h at 4 °C in 1× RIPA buffer (25 mM Tris-HCl, 150 mM NaCl, 1% Triton

X-100, 1% sodium deoxycholate and 0.1% sodium dodecyl sulfate) supplemented with protease and phosphatase inhibitor cocktails (Roche, 04693159001 and 04906837001). For probing endogenous ITGβ5, the samples were further boiled for 10 min at 95° C. After lysis, the samples were then spun for 15 min at 4° C to clarify lysates. The lysates were mixed with 2× Laemmli sample buffer (Bio-Rad, 1610747) and β-mercaptoethanol, boiled for 5 min at 95° C and subsequently subject to SDS–PAGE using a Mini-Protean vertical electrophoresis system (Bio-Rad, 1658026FC). After electrophoresis, the samples were transferred onto a nitrocellulose membrane using a Trans-Blot Turbo Transfer system (Bio-Rad, 1704150). The membranes were blocked with 3% BSA in 1× TBST buffer and then incubated with primary antibodies overnight at 4° C. The protein bands were visualized using HRP-conjugated secondary antibody and chemiluminescence under Azure Imaging Systems (Azure Biosystem).

## Co-immunoprecipitation

HEK293T cells (ATCC, CRL-3216) were transiently co-transfected with GFP–ITGβ5(715–769) or GFP–ITGβ5TM together with FCHo2_μHD–RFP or FCHo2_ΔμHD–RFP by electroporation. After 48 h of culture, cells were washed with ice-cold 1× PBS followed by exposure to freshly prepared 0.5 mM DSP (3,3′-dithiobis(succinimidyl propionate), Sigma-Aldrich) in 1× PBS for 30 min at RT. Cells were then incubated in an ice-cold 50 mM Tris-HCl buffer for 15 min and lysed in 1× RIPA buffer for 1 h at 4° C. The lysates were incubated with equilibrated GFP-Trap magnetic agarose (ChromoTek) overnight at 4° C. The beads were pelleted and washed with 1× RIPA buffer for 4 times. The pellets were subsequently resuspended in 2× Laemmli sample buffer (with β-mercaptoethanol) and boiled for 10 min at 95° C to elute and denature proteins. The samples were subject to SDS–PAGE and western blotting.

## Fluorescence imaging

Fluorescence images were acquired using an epi-fluorescence microscope (Leica DMI 6000B), controlled by MetaMorph software, with a ×100 (1.40 NA), ×60 (1.40 NA) or ×20 (0.80 NA) objective. The microscope is equipped with an ORCA-Flash4.0 Digital CMOS camera, a Lumencor SOLA light source and the following filter sets: 370-39/409/448-63 nm (blue emission), 484-25/505/524-32 nm (green emission), 560-32/581/607-40 nm (red emission) and 640-19/655/680-30 nm (far-red emission). We scanned nanoarrays left-to-right and top-to-down to identify cells. During live-cell imaging, cells were maintained in phenol-red-free DMEM (Gibco) supplied with 10% FBS at 37 °C with 5% of $CO_2$ in a stage top incubator (Tokai Hit, INUBSF-ZILCS). Ratiometric FRET imaging of live cells expressing talin tension sensors was performed using an epi-fluorescence microscope (Leica DMI 8000), controlled by Leica LAS X, with excitation at 496–516 nm from the Leica LED8 system. The donor emission at 527–551 nm and the acceptor emission at 602–680 nm were recorded using a K8 Scientific CMOS microscope camera. Confocal images were acquired using a Nikon A1R confocal microscope, controlled by Nikon NIS-Elements AR, with 1.2 Airy Unit using a ×20 (0.75 NA) or a ×40 (WD = 0.61 mm, 1.15 NA) water-immersion objective. The microscope is equipped with 405, 488, 561 and 633 nm lasers for excitation.

## Expansion microscopy imaging of the cell–nanopillar interface

Expansion microscopy was carried out as previously described[33]. In brief, following immunostaining, samples were incubated overnight at RT in a 1:100 dilution of AcX (Acryloyl-X, SE, 6-((acryloyl)amino) hexanoic acid succinimidyl ester, Invitrogen, A20770) in PBS. After washing with PBS, cells were incubated for 15 min in the gelation solution (19% (w/w) sodium acrylate, 10% (w/w) acrylamide, 0.1% (w/w) N, N′-methylenebisacrylamide) at RT. Then, nanochips were flipped cell side down onto a 70 μl drop of the gelation solution supplemented with

0.5% N,N,N′,N′-tetramethylenediamine and 0.5% ammonium persulfate on Parafilm and incubated at 37 °C for 1 h. After gelation, the nanochip with the hydrogel still attached was incubated in a 1:100 dilution of proteinase K in digestion buffer (50 mM Tris HCl (pH 8), 1 mM EDTA, 0.5% Triton X-100, 1 M NaCl) for 7 h at 37 °C. Hydrogels were then soaked twice in water for 30 min and then incubated overnight in water at 4 °C. To image samples, excess water on the hydrogels was carefully removed using a Kimwipe. Then, the hydrogels were mounted onto PLL-coated glass coverslips to prevent sliding during imaging.

## Super-resolution microscopy

Our method was adapted from previously described protocols[44,45]. The 3D single-molecule data were acquired using a custom-built wide-field double-helix point-spread function (PSF) inverted microscope. The samples were incubated in a blinking buffer solution containing 10% (w/v) glucose (BD Difco), 100 mM tri(hydroxymethyl) aminomethane-HCl (Thermo Fisher), 2 μl ml$^{-1}$ catalase, 560 μg ml$^{-1}$ glucose oxidase and 10 mM of cysteamine (all Sigma Aldrich), allowing for a low emitter concentration during data acquisition. The focus was first set to the coverslip using a bead as a reference. Standard localization-based approaches fitted the shape of the PSF of emitters to a 2D Gaussian to yield highly precise $xy$ positions. To extract the $z$ position, a double-helix phase mask was inserted in the Fourier plane of the microscope to modify the shape of the standard PSF to now have two lobes that rotate as a function of $z$. We fitted the double-helix PSF to two Gaussian functions and extracted the midpoint to estimate the $xy$ position. From the fit, we determined the lobe angle of the double-helix PSF. Then, using a carefully calibrated curve that relates the lobe angle to the $z$ position, we determined the $z$ position of the emitter. The localization precision was calculated from the detected photons using a formula calibrated specifically for our microscope. After processing, poorly localized emitters ($xy$ precision >20 nm or $z$ precision >40 nm or lobe distance >8 pixels) were removed from the reconstruction. Localizations were merged to correct for overcounting before quantification of any individual fibre diameters. The localized single-molecule positions were rendered using the Vutara SRX program (Bruker).

## SEM analysis

Nanostructures without cells were imaged by scanning electron microscopy (SEM; FEI Nova NanoSEM 450 and FEI Magellan 400 XHR). For SEM imaging of cells, the cells on nanostructures were fixed in 0.1 M sodium cacodylate buffer (pH 7.2) supplemented with 2% glutaraldehyde (Electron Microscopy Sciences) and 4% PFA overnight. The cells were then incubated in 0.1 M sodium cacodylate buffer supplied with 1% $OsO_4$ and 0.8% potassium ferricyanide (Electron Microscopy Sciences) for 2 h at 4 °C. The samples were dehydrated sequentially with 50, 70, 95, 100 and 100% (v/v) ethanol in water for 10 min each, and then dried by critical point drying. Before being imaged by SEM, the cell samples were sputter-coated with 3-nm-thick Cr.

## Quantitative analysis

All images were prepared and processed using ImageJ (Fiji). Quantifications were performed using ImageJ and MatLab (MathWorks) with custom codes.

**Quantification of nanostructure sizes.** The following parameters were measured in Fiji by counting pixels: the height, width and length of bars; the height, diameters (top, middle and bottom) of nanopillars. If the samples were tilted 45°, a correction factor was applied to the $z$ dimension. The diameters of curvatures at bar ends were measured using the Kappa curvature analysis plugin (open source) in Fiji.

**Quantification of the nanobar end/side ratio.** As nanobars are regularly spaced, the locations of nanobars were automatically detected from the bright-field images using a custom MatLab program, which

generates an array of evenly spaced square masks with nanobars at the centre. These square masks were averaged to generate a smooth and averaged image of the nanobar. This averaged image was then used to identify the bar-end and the bar-side locations with respect to the averaged square mask. Then, these relative locations were translated to individual square masks to identify the bar-end and the bar-side locations for individual nanobars. These regions of interest (ROIs) were then applied to the integrin or membrane colour channels.

For each nanobar, the end/side ratio of ITGβ5 was calculated and then normalized by dividing it with the end/side ratio of CaaX, except in Fig. 3i, for which the membrane is not labelled and the end/side ratio of ITGβ5 was directly calculated. The end/side ratios of individual nanobars in a cell were averaged and are reported as a single data point. A detailed step-by-step description of the quantification of the nanobar end/side ratio with examples has been provided in a previous study[46]. For cells on gradient bars, the normalized end/side ratios of the same size nanobars in each imaging field were averaged and are reported as a single data point. This is because very few bars with the same diameter were covered by a single cell.

**Quantification of the ITGβ5/membrane ratios on nanopillars.** Nanopillars are also regularly spaced like nanobars. The square masks with nanopillars at the centre were generated by using the same custom MatLab code. For nanopillars, we used CaaX or CellMask membrane images instead of bright-field images to generate the square masks as the membrane images had better qualities. Based on the averaged square mask of the plasma membrane channel within each cell, the ROIs for 'at nanopillar' and 'flat region surrounding nanopillar' were respectively defined as 0–9 and 10–20 pixel from the centre of a nanopillar. These ROIs were then applied to each mask region in both integrin and membrane channels. Mean intensities in each ROI were measured. The ITGβ5/membrane ratio for each ROI was calculated and normalized by the mean of ITGβ5/membrane ratio averaged over the entire cell and is reported as a single data point. The normalized ITGβ5/membrane ratios of individual ROIs are reported.

**Quantification of FRET ratios.** The method to quantify FRET ratios was adapted from previous studies[28,29]. The fluorescent images of the donor (Ypet) were autothresholded using the 'Moments' method in Fiji to generate binary maps for all adhesions. The bright-field images and the custom MatLab code were used to generate binary location maps for nanopillars, in which the ROI of a nanopillar was defined as 0–9 pixels to the nanopillar centre. The binary location map of nanopillars was multiplied by that of all adhesions to generate the mask for curved adhesions. The rest adhesion ROIs were defined as focal adhesions (clathrin-containing adhesions do not contain talin-1). The (acceptor emission intensity/donor emission intensity) ratios (FRET ratios) of curved adhesions and focal adhesions were calculated and then normalized by the FRET ratios averaged over the entire cell. The normalized FRET ratios were averaged for each cell and are reported as a single data point. For each cell, the FRET ratios at adhesions were normalized by the mean FRET ratio across the whole cell to minimize the cell-to-cell variation.

**Spearman's correlation coefficient at nanopillars.** To analyse the correlation between a protein of interest (POI) and the ITGβ5/membrane ratio at nanopillars, the location masks for nanopillars were first generated as described above. The means of POI, ITGβ5 and membrane (cell surface SNAP-tag AF647) at each nanopillar (0–9 pixels to the nanopillar centre) were measured. For each cell, the ITGβ5/membrane ratios at individual nanopillars were calculated and normalized by the mean ITGβ5/membrane ratio across the whole cell. From two arrays of POI intensities and ITGβ5/membrane ratios at individual nanopillars, their Spearman's correlation (the nonparametric version of the Pearson's correlation) in each cell was calculated using Prism 9.

**Dynamic measurements of F-actin and FCHo2 fluctuations at nanopillars.** The location masks for nanopillars were generated as described above. For cells on gelatin, ITGβ5 appears uniform on the cell membrane and across all nanopillars. The intensities of LifeAct at all nanopillars (0–9 pixels to the nanopillar centres) in a time-lapse image stack were measured. For cells on vitronectin, the intensities of ITGβ5 at individual nanopillars were first quantified. For F-actin dynamic measurements, nanopillars with the top 25% and bottom 25% ITGβ5 intensities were grouped as high-β5 and low-β5 nanopillars, respectively. For FCHo2 dynamic measurements, nanopillars for each cell were divided into two groups based on ITGβ5 intensities: high-β5 nanopillars (top 25%) and low-β5 nanopillars (bottom 75%). The FCHo2 intensities in time series were normalized by their values at time 0. The temporal standard deviations of LifeAct or normalized FCHo2 intensities at individual nanopillars are reported.

**Quantification of curved adhesions and focal adhesions for 2D flat surfaces and 3D ECMs.** For quantifications of the number and the area of curved adhesions and focal adhesions in 2D, two-channel fluorescence images (anti-ITGβ5 and anti-vinculin, or anti-ITGβ5 and FCHo2-GFP) were separately autothresholded using the 'Moments' method in Fiji. The resulting binary image of ITGβ5 was multiplied by the binary image of FCHo2 to generate overlapping ROIs for curved adhesions. Similarly, the ITGβ5 binary image was multiplied by the vinculin binary image to generate overlapping ROIs for curved adhesions. To eliminate noise and artefacts, the overlapping ROIs were then analysed with a low threshold of 10 pixels in Fiji using the 'analyze-particle' tool. Then, the number and the average size (area) of remaining ROIs for each cell was automatically quantified using the 'analyze-particle' tool. The analysis was automatically processed with the same threshold and was carried out to ensure unbiased comparison between curved adhesion and focal adhesions. Therefore, the analysis classified occasional ITGβ5–FCHo2 colocalizations on flat surfaces as curved adhesions, even though they are probably random events and not real curved adhesions.

For quantifications of the number and the volume of curved adhesions and focal adhesions in 3D, two-channel confocal 3D-stack images were autothresholded to remove diffusive background and segmented using the 3D interactive 'thresholding segmentation' tool of the 3Dsuit plug-in in Fiji. This step was used to identify protein accumulation. The resulting binary 3D stack of ITGβ5 was multiplied by the binary 3D stack of FCHo2 to generate 3D overlapping ROIs for the quantification of curved adhesions. Similarly, the binary 3D stack of ITGβ5 was multiplied by the binary 3D stack of vinculin to generate 3D overlapping ROIs for focal adhesions. Subsequently, the number and the average volume of adhesions for each cell was measured using the '3D objects counter' tool in Fiji with a low threshold of 25 voxels. Similar to 2D analysis, the analysis was carried out with the same parameter applied for both sets of data to avoid bias.

**Quantification of the cell depth in 3D matrices.** Confocal 3D-stack images of GFP–CaaX were used to identify cell positions. The first step involved the application of the '3D edge filter' of the 3Dsuit plug-in in Fiji to the GFP–CaaX 3D stack to identify the cell boundaries. The second step involved segmenting the 3D cell boundary stack using Fiji Macro 3D ART VeSElecT to identify 3D ROIs for individual cells. For this step, a low volume threshold of 500 voxels was applied to remove cell debris and extracellular vesicles. The third step calculated the centre coordinates $(x, y, z)$ of each cell ROIs using the '3D objects counter' tool in Fiji.

To determine the depth for each cell, we first calculated the surface height position. Confocal 3D-stack images of AF647-labelled-collagen were subjected to the surface peeler Fiji macro to identify the surface of the 3D matrix. In this process, a Gaussian blurring filter (radius of 50 pixels) was applied to fill the gaps between fibres. This step

identified a *z*-height value for each (*x*, *y*) position. Then, the depth of a cell was calculated by subtracting the *z*-position of the cell centre with the *z*-position of the surface at the (*x*, *y*) location of cell centres. All cells in 3D stacks were pooled together to generate the cell depth distribution.

## Statistics and reproducibility
All data are displayed and were statistically analysed using Prism 9 (GraphPad Software). The experiments were not randomized. All experiments were independently repeated two times, unless specified otherwise in the legends, with similar results obtained to ensure reproducibility. No statistical method was used to predetermine sample sizes, which were determined on the basis of previous studies in the field. The normal distribution of each sample group was verified using the D'Agostino–Pearson normality test ($\alpha = 0.05$). For parametric datasets, we used unpaired two-tailed Welch's *t*-test to compare two groups, and one-way analysis of variance (ANOVA) with Tukey's or Bonferroni's multiple comparison to compare more than two groups. For nonparametric datasets, we used Mann–Whitney test to compare two groups, and Kruskal–Wallis test with Dunn's multiple comparison to compare more than two groups. The exact *P* values and sample sizes (*n*) are denoted in the figures and figure legends. Significance was considered at $P < 0.05$. The test statistics, including confidence levels and degrees of freedom, are provided in Supplementary Table 5. No data were excluded from the tests. The investigators were not blinded to allocation during experiments and outcome assessment.

## Reporting summary
Further information on research design is available in the Nature Portfolio Reporting Summary linked to this article.

## Data availability
Plasmids originally generated in this study and their sequence information are available at Addgene (identifiers 205090, 205091, 205092, 205093, 205094 and 205095). The expression level of integrins in U2OS cells was obtained from the Human Protein Atlas project's RNA HPA cell line gene data available from https://www.proteinatlas.org (v.23.0). Source data are provided with this paper. All other data supporting the findings of this study are available from the corresponding author upon reasonable request.

## Code availability
Custom MatLab codes used for analyses are available at GitHub (https://github.com/wzhang5publication/Data-analysis). Any additional information will be available from the corresponding author upon reasonable request.

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

## Acknowledgements
This work was supported by NIH grants (R35GM141598, R01HL165491, R01NS121934), an Ono Pharma Breakthrough Initiative Award, and a Packard Fellowship for Science and Engineering (to B.C). The nanofabrication in this work was performed at the Stanford Nanofabrication Facility under the NSF National Nanotechnology Coordinated Infrastructure programme and at the Stanford Nano Shared Facilities supported by the National Science Foundation under award ECCS-2026822. The authors were also supported by a Stanford University Center for Molecular Analysis and Design fellowship (to C.-H.L.), a National Institutes of Health grant Biotechnology Training Grant fellowship (to M.L.N.), and Molecular Biophysics Training Program T32 GM136568 (to C.E.L.). We thank T. Jones for insightful discussions, S. Dixon for the lung fibroblast cells, W. E. Moerner for assistance with 3D super-resolution imaging and C. Bertozzi for the confocal microscope.

## Author contributions
W.Z. and B.C. conceived the study and designed the experiments. C.-T.T., X.L. and Z.J. fabricated the nanostructures and performed SEM measurements. W.Z., C.-H.L., Y.Y., C.E.L. and M.L.N. constructed the plasmids. Y.Y. and B.C. developed MatLab codes for data analyses. W.Z., C.-H.L. and C.E.L. performed most of the cell experiments and fluorescence microscopy. M.L.N. performed expansion microscopy. W.Z., C.-H.L. and C.-T.T. analysed most of the data. A.R.R. carried out super-resolution imaging, reconstruction and analyses. W.Z., C.-H.L., M.L.N., C.E.L. and B.C. wrote the paper. All authors discussed the results and contributed to the paper.

## Competing interests
The authors declare no competing interests.

## Additional information
**Extended data** is available for this paper at https://doi.org/10.1038/s41556-023-01238-1.

**Correspondence and requests for materials** should be addressed to Bianxiao Cui.

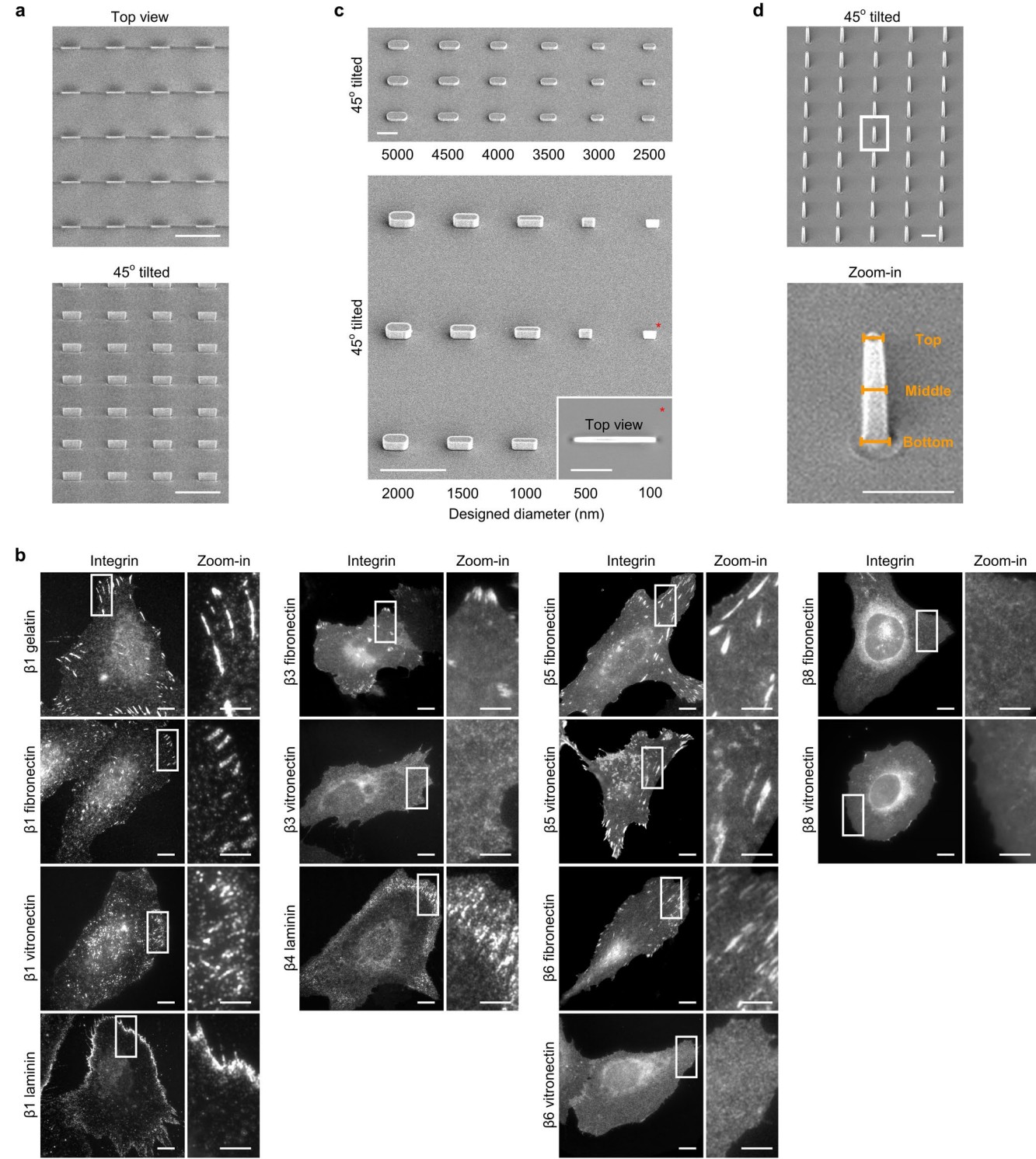

**Extended Data Fig. 1 | SEM characterization of nanostructures and integrin activation on ECM protein-coated substrates. a**, Representative SEM images show a 200 nm-wide vertical nanobar array. The same sample was viewed either from the top (top) or at a 45° angle (bottom). Scale: 5 µm. **b**, Fluorescence images of integrin β subunits on flat surfaces coated with different ECM proteins (gelatin, laminin, fibronectin or vitronectin). Endogenous β1 was immunolabelled, while β3, β4, β5, β6, and β8 were tagged with GFP and transiently expressed. Scale: full-size, 10 µm; insets, 5 µm. **c**, Representative SEM images of a gradient bar

array with designed end curvature diameters ranging from 100 to 5000 nm. The sample was tilted 45°. Scale: 10 µm. Inset: the top view of a 100 nm-wide nanobar. Scale, 1 µm. **d**, Representative SEM images show a vertical nanopillar array. The sample was tilted 45°. The zoom-in image shows a single nanopillar. The top, middle, and bottom diameters of each nanopillar were measured. Scale: full-size and inset, 1 µm. Statistical analysis of the geometrical dimensions of the nanostructures (in **a**, **c**, **d**) is provided in Supplementary Table 1.

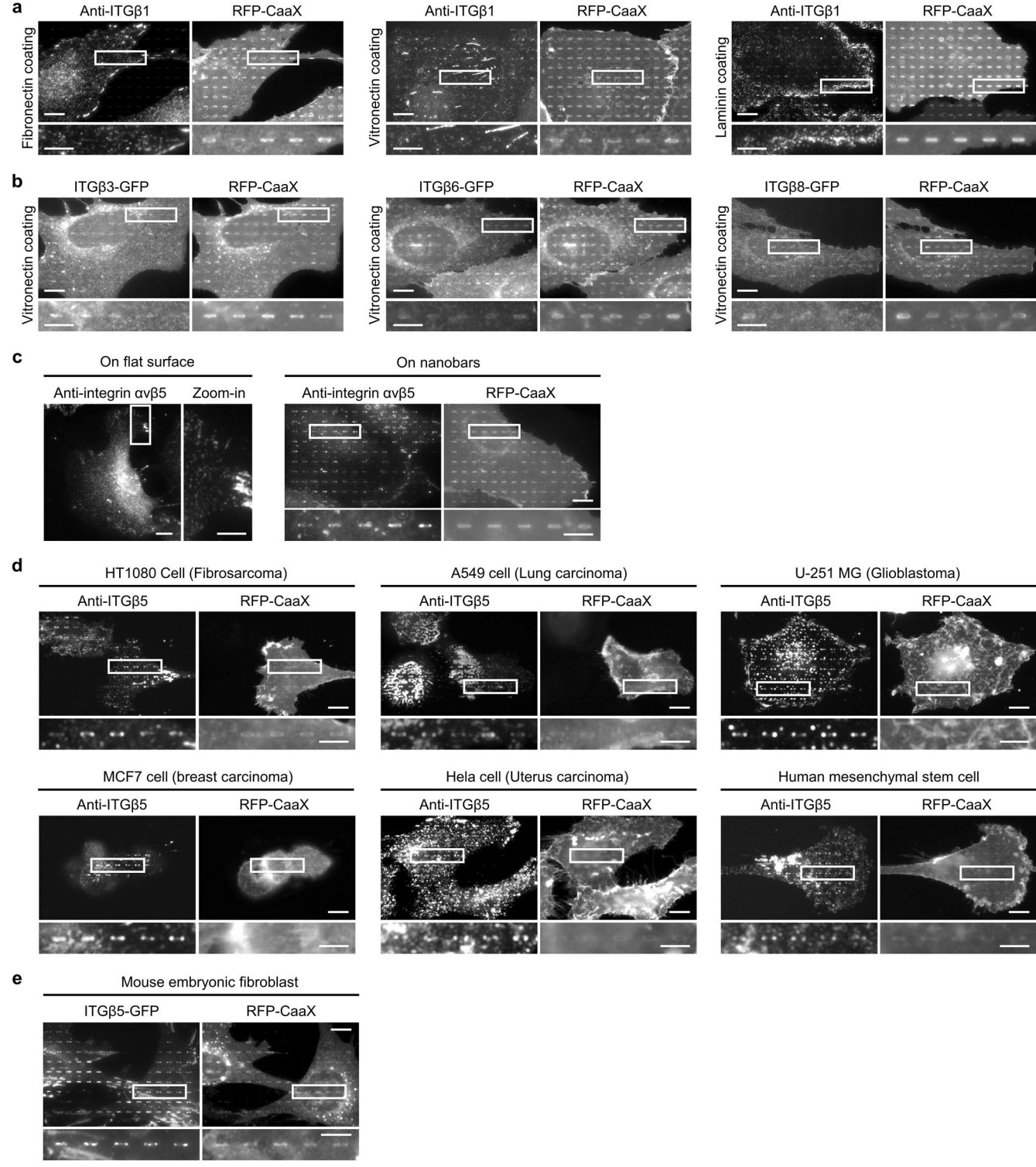

**Extended Data Fig. 2 | Positive membrane curvature induces the preferential accumulation of integrin β5 but not other integrin β isoforms. a** and **b**, Fluorescence images of endogenous integrin β1 (**a**) and transiently expressed GFP-tagged integrin β3, β6, and β8 (**b**) in U2OS cells expressing a plasma membrane marker RFP-CaaX on 200-nm nanobar arrays coated with different ECM proteins (fibronectin, vitronectin or laminin). All of them show no preference for the nanobar ends. Quantifications of their curvature preferences are presented in Fig. 1i. Scale: full-size, 10 μm; insets, 5 μm. **c**, Fluorescence images showing that anti-αvβ5 in U2OS cells localizes to focal adhesions on

vitronectin-coated flat surfaces (left), and preferentially accumulates at the ends of vitronectin-coated nanobars in U2OS cells expressing RFP-CaaX (right). Scale: full-size, 10 μm; insets, 5 μm. **d**, Fluorescence images showing that anti-ITGβ5 preferentially accumulates at the ends of vitronectin-coated nanobars in HT1080, A549, U-251 MG, MCF7, HeLa, and human mesenchymal stem cells. Scale: full-size, 10 μm; insets, 5 μm. **e**, Fluorescence images showing that ITGβ5-GFP preferentially accumulates at the ends of vitronectin-coated nanobars in mouse embryonic fibroblasts. Scale: full-size, 10 μm; insets, 5 μm.

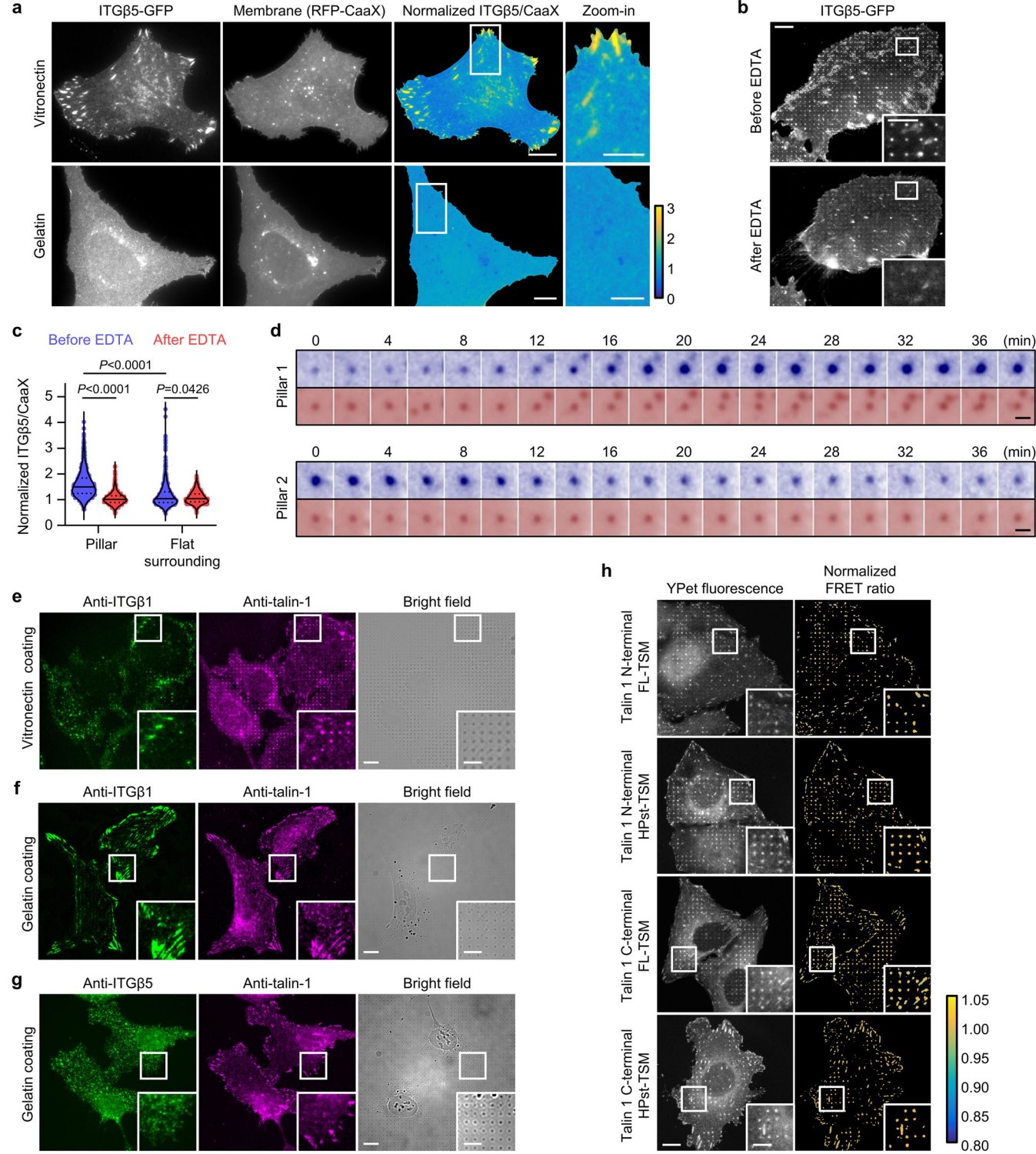

**Extended Data Fig. 3 | See next page for caption.**

**Extended Data Fig. 3 | Integrin β5 preferentially accumulates at nanopillars in a ligand-dependent manner and recruits talin-1. a**, Fluorescence images of ITGβ5-GFP and plasma membrane marker RFP-CaaX on vitronectin-coated (top) and gelatin-coated (bottom) flat surfaces. Ratiometric images of ITGβ5/ membrane (normalized by its mean per cell) are shown in the Parula colour scale. Scale: full-size, 10 μm; insets, 5 μm. **b**, Ethylenediaminetetraacetic acid (EDTA) treatment induces a dramatic reduction of ITGβ5-GFP accumulation at nanopillars in a live U2OS cell. Scale: full-size, 10 μm; insets, 5 μm. **c**, Quantification of the normalized ITGβ5/CaaX ratio at nanopillars and their surrounding flat regions. n = 994/711/994/711 nanopillars and their surrounding flat regions, from three independent cells. Medians (lines) and quartiles (dotted lines) are shown. *P* values calculated using paired Kruskal-Wallis test with Dunn's multiple-comparison. **d**, Time-lapsed images of ITGβ5-GFP and CellMask Orange at Pillar 1 and 2 in Fig. 2e. At Pillar 1, the curved adhesion gradually

assembles. At Pillar 2, the curved adhesion gradually disassembles. Scale: 1 μm. **e**, Fluorescence and bright-field images showing that immunolabelled ITGβ1 does not accumulate at vitronectin-coated nanopillars, while immunolabelled talin-1 does. Scale: full-size, 10 μm; insets, 5 μm. **f** and **g**, Fluorescence and bright-field images of immunolabelled talin-1 together with immunolabelled ITGβ1 (**f**) or ITGβ5 (**g**) on gelatin-coated nanopillar substrates. Talin-1 colocalizes with ITGβ1 in focal adhesions but does not accumulate at nanopillars. ITGβ5 appears mostly diffusive on gelatin-coated substrates. These results indicate that nanopillar-induced talin accumulation depends on ITGBβ5 and its ligands. Scale: full-size, 10 μm; insets, 5 μm. **h**, Fluorescence (YPet) and normalized ratiometric FRET images (in Parula colour scale) of the talin tension sensor N-terminal and C-terminal controls. Scale: 10 μm; insets, 5 μm. Source numerical data are available in source data.

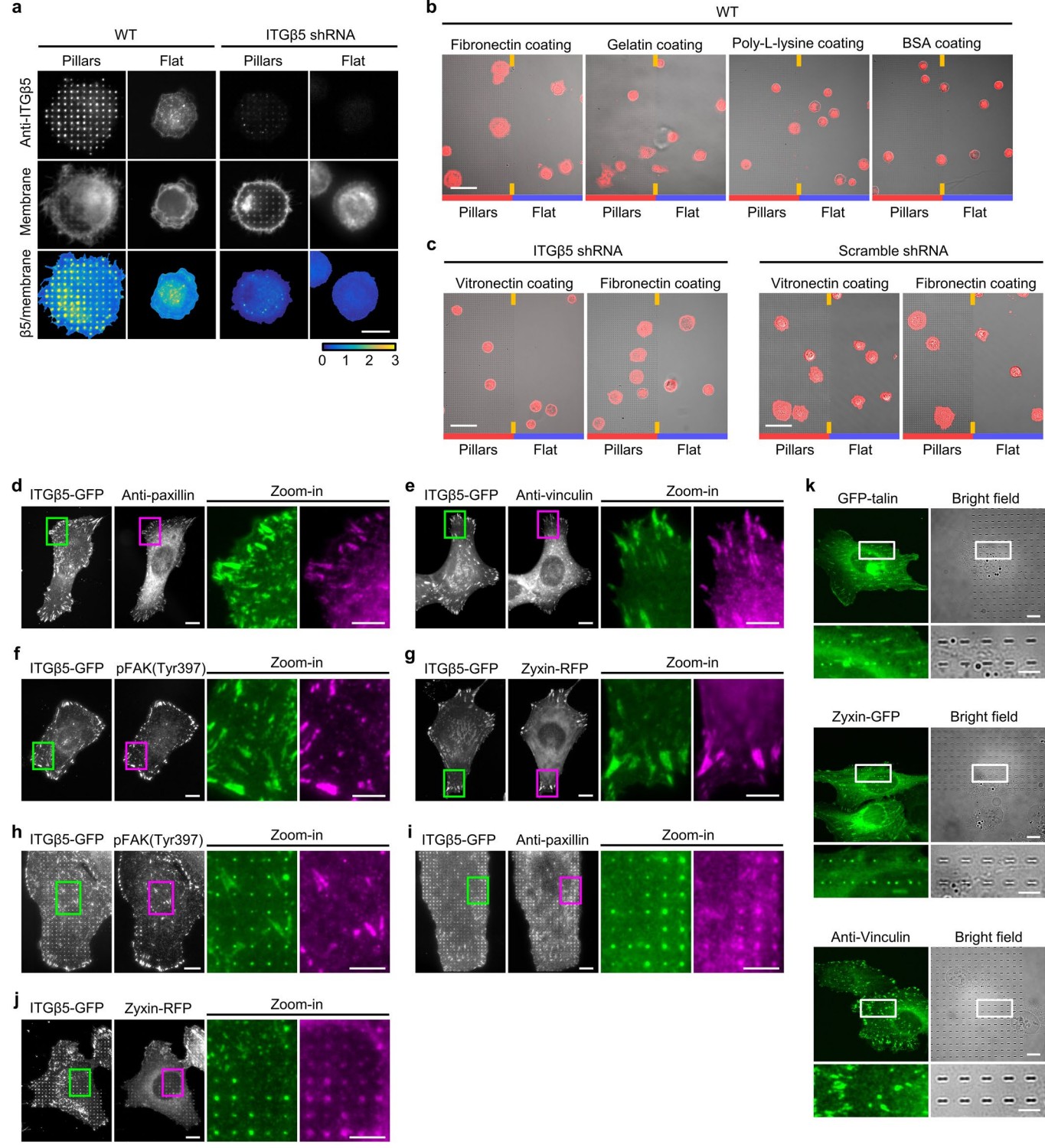

**Extended Data Fig. 4 | See next page for caption.**

**Extended Data Fig. 4 | Curved adhesions promote early-stage cell spreading and involve a subset of focal adhesion proteins. a**, ITGβ5 accumulates at vitronectin-coated nanopillars within 30 min after seeding. Cells spread to a larger area on nanopillars compared to on flat areas on the same substrate. Knockdown of ITGβ5 with shRNAs largely abolished the nanopillar-induced early cell spreading. Scale: 10 μm. **b**, Overlay of bright-field and fluorescence images of wild type (WT) U2OS cells stably expressing GFP-CaaX on both nanopillars and flat surfaces in the same imaging fields. The cells were seeded on substrates and cultured for 30 min before imaging. Scale: 50 μm. **c**, Overlay of bright-field and fluorescence images of U2OS cells transduced with ITGβ5 or scramble shRNAs and stably expressing GFP-CaaX on both nanopillar arrays and flat surfaces in the same imaging fields. The cells were seeded on substrates and cultured for 30 min before imaging. Scale: 50 μm. **d** to **g**, Fluorescence images showing that ITGβ5-GFP colocalizes with focal adhesion proteins paxillin (**d**), vinculin

(**e**), pFAK (Tyr397) (**f**), and zyxin (**g**) at focal adhesion-like patches in U2OS cells on vitronectin-coated flat surfaces. Scale: full-size, 10 μm; insets, 5 μm. **h** to **j**, Representative fluorescence images showing that ITGβ5-GFP accumulation at vitronectin-coated nanopillars spatially correlates with paxillin (**i**) and zyxin (**j**), but not with pFAK (Tyr397) (**h**) in U2OS cells. The cell membrane was marked with transiently expressed surface SNAP-tag conjugated with AF647. The normalized ITGβ5/membrane was used to measure ITGβ5 accumulation for the correlation quantification in Fig. 3b. Scale: full-size, 10 μm; insets, 5 μm. Endogenous paxillin, vinculin, and pFAK (Tyr397) were immunolabelled. Zyxin was tagged with RFP and transiently expressed (**d-j**). **k**, Fluorescence images of cells cultured on nanobars showing that GFP-talin and zyxin-GFP preferentially accumulate at the nanobar ends. However, anti-vinculin shows no accumulation at nanobar locations. Scale: full-size, 10 μm; insets, 5 μm.

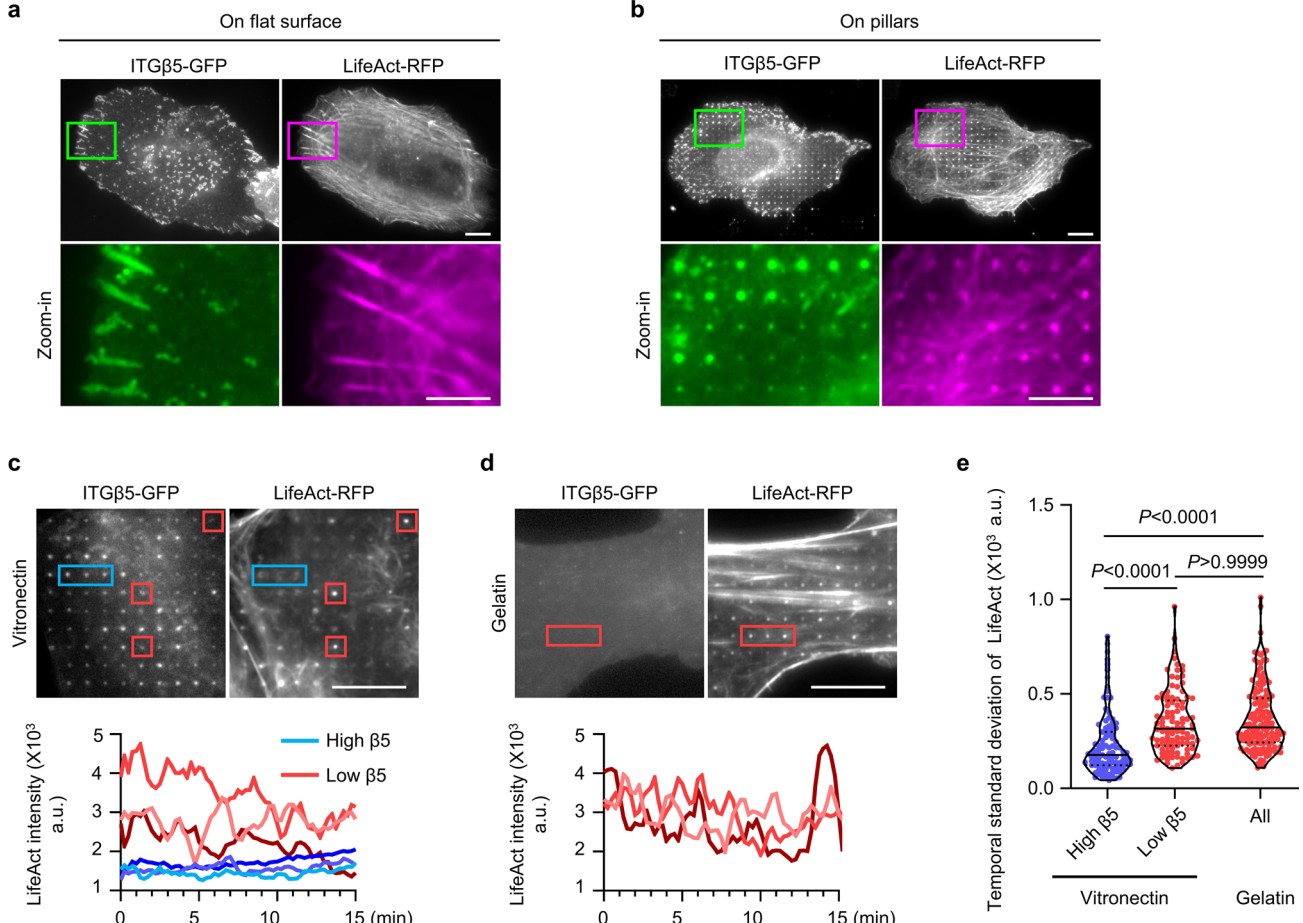

**Extended Data Fig. 5 | Curved adhesions are linked to a stable population of actin filaments at nanopillars. a**, On flat areas, actin stress fibres (F-actin labelled by LifeAct-RFP) are anchored to ITGβ5-marked focal adhesion patches. Scale: full-size, 10 μm; insets, 5 μm. **b**, On nanopillar areas, both ITGβ5 and F-actin accumulate at nanopillars. However, nanopillars with high β5 accumulations usually do not have high levels of F-actin. A previous study shows that the membrane curvature around gelatin-coated nanopillars induces actin accumulation, but the accumulation is highly dynamic with a lifetime of 1-2 minutes[31]. As curved adhesions are stable and do not form on gelatin-coated nanopillars, we hypothesize that curved adhesions involve a different population of F-actin. Scale: full-size, 10 μm; insets, 5 μm. **c**, Dynamic correlation between ITGβ5-GFP and LifeAct-RFP in cells cultured on vitronectin-coated nanopillars. There are two distinct populations of the F-actin that accumulate at nanopillars: a stable population of F-actin at high-β5 nanopillars, and a dynamic population

F-actin at low-β5 nanopillars. The F-actin intensity of the dynamic population is usually brighter than that of the stable population. Scale: 10 μm. **d**, On gelatin-coated nanopillars that do not induce β5 accumulation, only the highly dynamic population of F-actin was observed at nanopillars. Scale: 10 μm. **e**, Quantifications of time-dependent fluctuation confirm two F-actin populations: a stable population on high-β5 nanopillars and a dynamic population at low-β5 nanopillars. Nanopillars with the top 25% and bottom 25% ITGβ5 intensities were grouped as high-β5 and low-β5 nanopillars, respectively. The dynamic population of F-actin on vitronectin-coated nanopillars is similar to the dynamic F-actin on gelatin-coated nanopillars. Therefore, curved adhesions involve a stable subpopulation of F-actin at nanopillars. n = 103/103/153 nanopillars, from two independent cells. Medians (lines) and quartiles (dotted lines) are shown. *P* values calculated using Kruskal-Wallis test with Dunn's multiple-comparison. Source numerical data are available in source data.

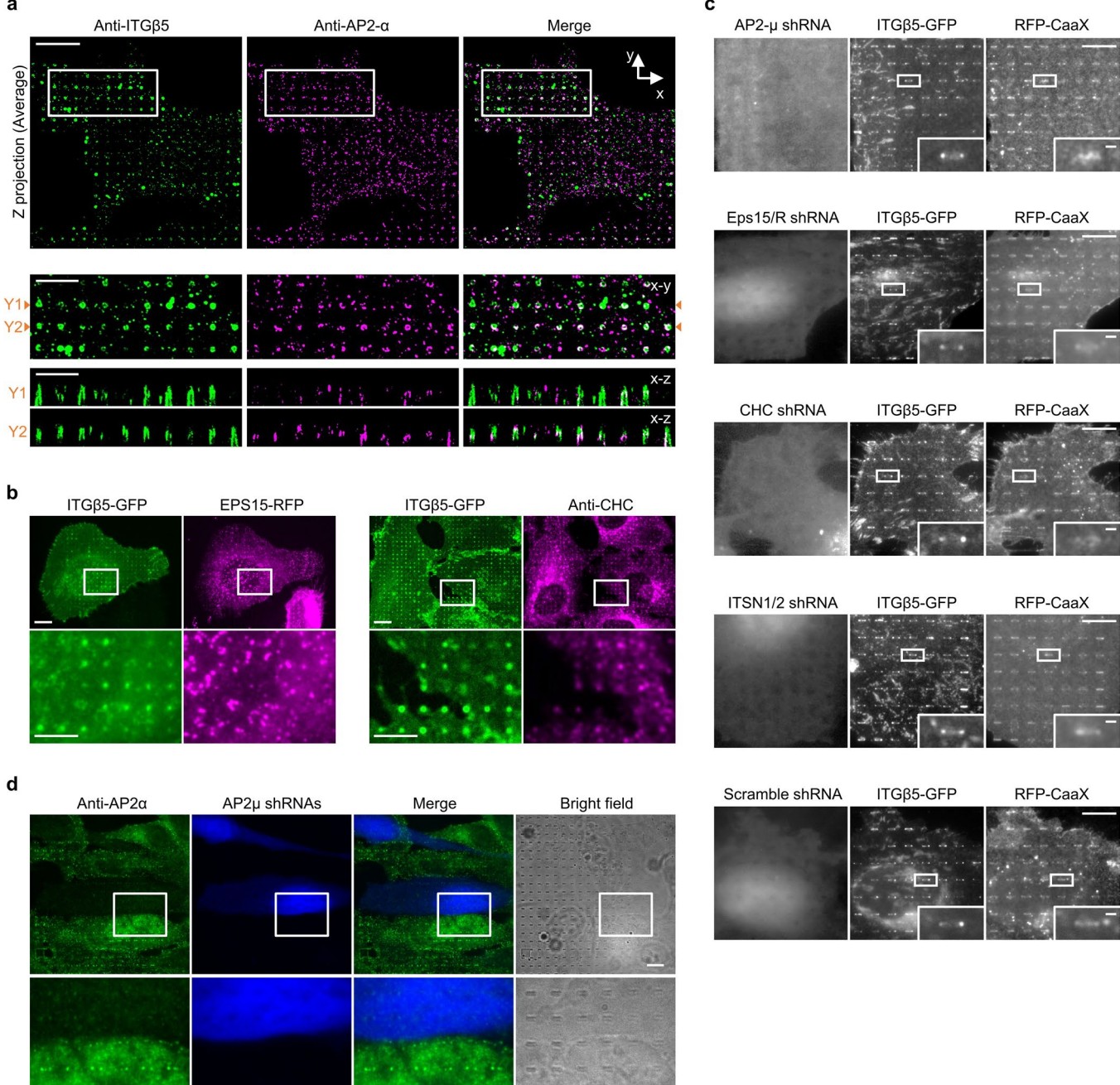

**Extended Data Fig. 6 | Curved adhesions are different from clathrin lattices.**
**a**, Expansion microscopy showing that ITGβ5 and AP2 are not spatially correlated at the nanopillar-membrane interface. Expansion microscopy is used to increase the spatial resolution of optical imaging for the nanopillar-membrane interface (see the method section and Ref. 33 for detailed descriptions). Both ITGβ5 and AP2-α are immunolabelled. Top: x-y images of the z projection (average). Bottom: zoom-in x-y images of the area indicated by the white boxes. x-z images showing the distribution of immunolabelled ITGβ5 and AP2-α along nanopillars at y = Y1 and y = Y2 in the zoom-in images. Even when ITGβ5 and AP2-α accumulate on the same nanopillar in the x-y image, they are not correlated in the z-dimension. Scale: full-size, 10 μm; insets, 5 μm. **b**, Right: Both ITGβ5-GFP and immunolabelled clathrin heavy chain (CHC) accumulate at vitronectin-coated nanopillars, but their intensities are not correlated. Nanopillars with high intensities of ITGβ5-GFP are usually not the nanopillars with high intensities of

anti-CHC. Left: ITGβ5-GFP accumulates at vitronectin-coated nanopillars, but the co-transfected EPS15-RFP does not show strong accumulation or correlation with ITGβ5-GFP at these nanopillars. Scale: full-size, 10 μm; insets, 5 μm. **c**, Representative fluorescence images showing that the shRNA knockdown of AP2-μ, clathrin heavy chain (CHC), EPS15/R, or ITSN1/2 does not affect the accumulation of ITGβ5-GFP accumulation at the ends of vitronectin-coated nanobars in U2OS cells expressing RFP-CaaX membrane marker. BFP expression is a marker of shRNA transfection. Quantifications of the β5-GFP curvature preference under these conditions are presented in Fig. 3d. Scale: full-size, 10 μm; insets, 1 μm. **d**, Validation of AP2 knockdown by immunofluorescence. Immunofluorescence showing that the transfection of AP2-μ shRNAs (indicated by BFP expression) can reduce the appearance of AP2 complexes and the accumulation of AP2 complex at the ends of vitronectin-coated nanobars in U2OS cells. Scale: full-size, 10 μm; insets, 5 μm.

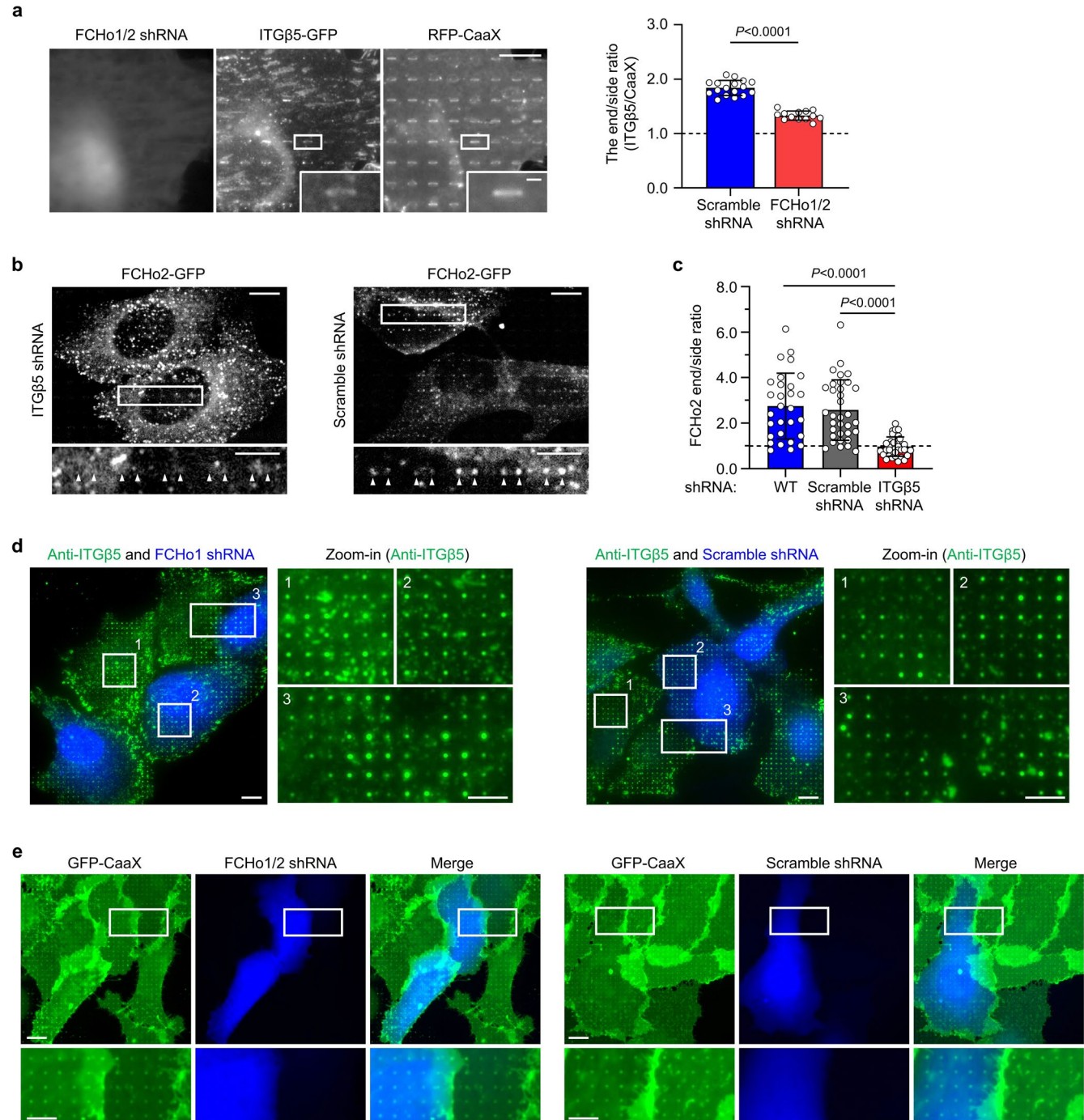

**Extended Data Fig. 7 | FCHo2, but not FCHo1, is correlated with ITGβ5 in curved adhesions. a**, Left: Representative fluorescence images showing that knockdown of FCHo1/2 reduces the β5-GFP accumulation at vitronectin-coated nanobar ends. Scale: full-size, 10 μm; insets, 1 μm. Right: Quantifications of the β5-GFP curvature preference in U2OS cells transfected with scramble or FCHo1/2 shRNAs. n = 17/13 cells, from two independent experiments. Data are presented as the mean ± s.d. *P* values calculated using two-tailed t-test. **b**, Representative fluorescence images showing that compared with scramble shRNA transfection (right), shRNA knockdown of ITGβ5 (left) reduces the FCHo2-GFP accumulation at the ends of vitronectin-coated nanobars in U2OS cells. Scale: full-size, 10 μm; insets, 5 μm. **c**, Quantifications of the FCHo2-GFP curvature preference in wild-type, scramble shRNA-transfected, or ITGβ5-knockdown U2OS cells.

n = 29/34/36 cells, from two independent experiments. Data are presented as the mean ± s.d. *P* values calculated using one-way ANOVA with Tukey's multiple comparison. **d**, Representative fluorescence images showing that neither shRNA knockdown of FCHo1 (left) nor scramble shRNA transfection (right) affects ITGβ5 accumulation at vitronectin-coated nanopillars. Scale: full-size, 10 μm; insets, 5 μm. **e**, Representative fluorescence images showing that compared with scramble shRNA transfection (right), shRNA knockdown of FCHo1/2 (left) does not affect membrane wrapping around vitronectin-coated nanopillars in U2OS cells expressing GFP-CaaX membrane marker. BFP expression is a marker of shRNA transfection. Scale: full-size, 10 μm; insets, 5 μm. Source numerical data are available in source data.

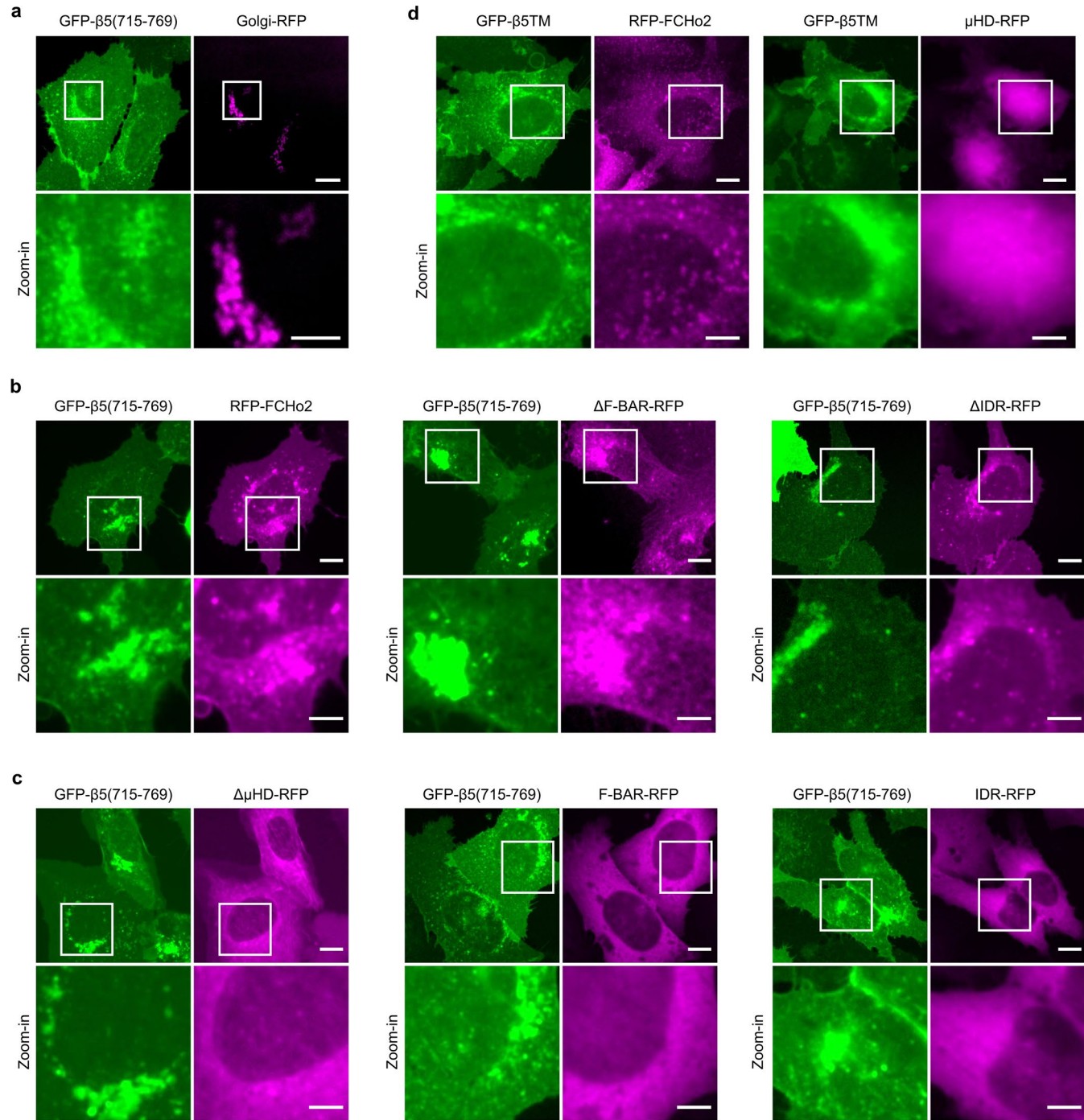

**Extended Data Fig. 8 | GFP-β5(715-769) induces dramatic redistribution of µHD-containing FCHo2 variants to the plasma membrane and the Golgi apparatus. a**, GFP-β5(715-769) is located on the plasma membrane with some accumulations around the perinuclear region that colocalizes with Golgi apparatus marker Golgi-RFP. Scale: full-size, 10 µm; insets, 5 µm. **b**, When co-expressed with GFP-β5(715-769), three µHD domain-containing FCHo2 variants (RFP-FCHo2, FCHo2_ΔF-BAR-RFP, and FCHo2_ΔIDR-RFP) are redistributed to the plasma membrane and colocalize with GFP-β5(715-769) in the perinuclear region. Scale: full-size, 10 µm; insets, 5 µm. **c**, When co-expressed with GFP-β5(715-769), three FCHo2 variants that don't contain µHD domain (FCHo2_ΔµHD-RFP, FCHo2_F-BAR-RFP and FCHo2_IDR-RFP) are highly diffusive in the cytosol. **d**, When co-expressed with the negative control GFP-β5TM, RFP-FCHo2 shows cytosolic diffusive pattern with small puncta (left) and FCHo2_µHD is highly cytosolic (right and in Fig. 4l). Scale: full-size, 10 µm; insets, 5 µm.

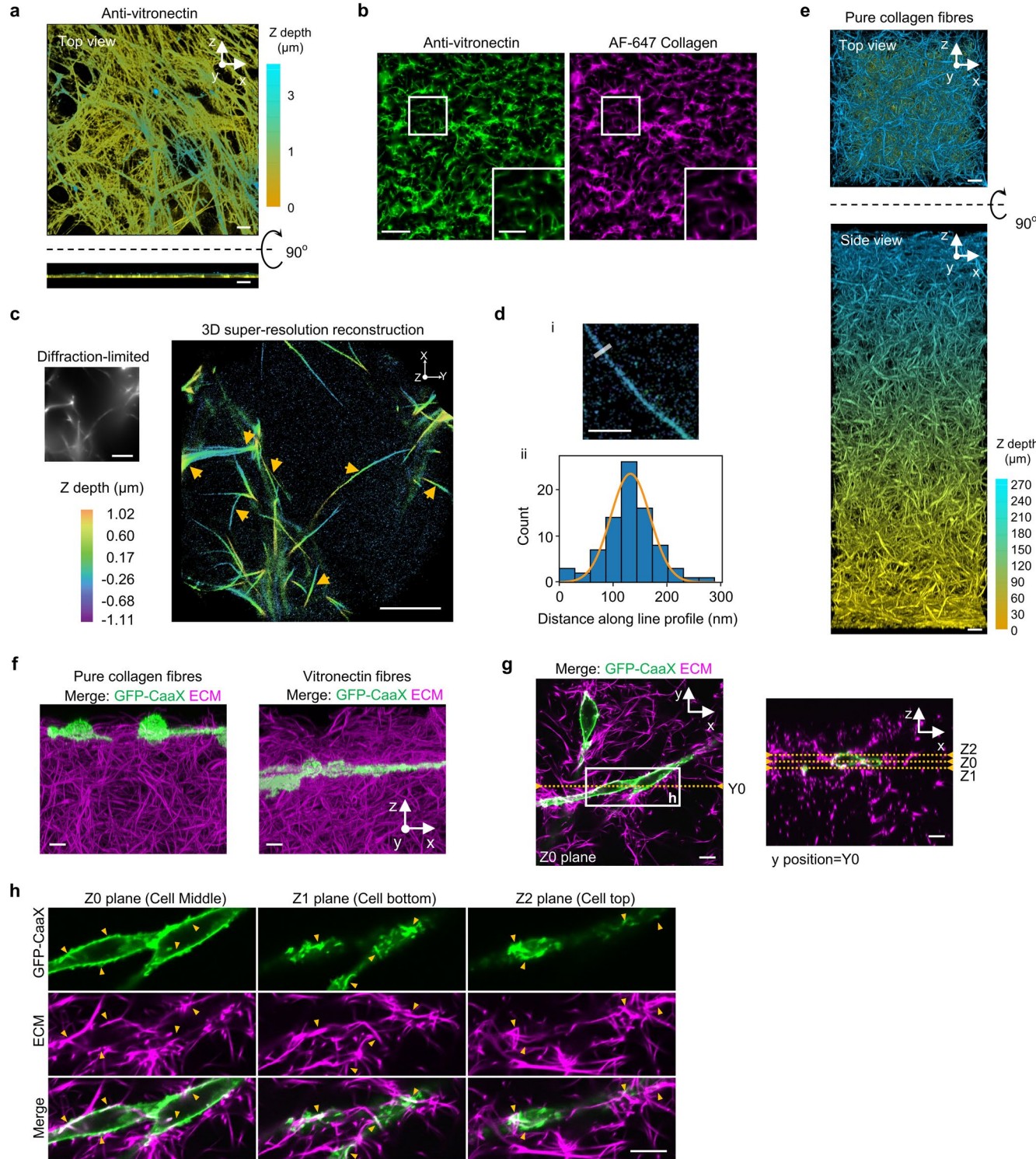

**Extended Data Fig. 9 | Cell plasma membranes are deformed by 3D ECM fibres. a**, Anti-vitronectin staining illustrates the fibre morphology of IMR-90 lung fibroblast-derived ECM. Z-depth is colour-coded. Scale: 10 µm. **b**, In 3D ECM made of vitronectin fibres, colocalization of AF647-collagen with immunolabelled vitronectin confirmed the incorporation of vitronectin in ECMs. Scale: full-size, 50 µm; insets, 25 µm. **c**, 3D super-resolution reconstruction of AF647-labelled collagen fibres. Top left: the widefield diffraction-limited image. Right: the z projection of the 3D super-resolution reconstruction. Yellow arrows point to some individual collagen fibres. Colour encodes the z position. Scale: 5 µm. **d**, Extracting the diameter of individual fibres. (i) The image shows an example of a resolved individual fibre. Transverse line profiles are taken across the fibre to estimate its diameter. (ii) Localizations from the line profile are

binned into a histogram which is then fit to a Gaussian function (orange line). The full width at half maximum is extracted to estimate the diameter. Scale: 1 µm. **e**, Top and side views (3D projection) of a thick 3D ECM made of AF647-labelled pure collagen fibres. Z depth is colour coded. Scale: 10 µm. **f**, Representative 3D images (x-z projection) of U2OS cells expressing GFP-CaaX, 72 hrs after being plated on the top of a matrix made of pure collagen fibres (left) or vitronectin fibres (right). Scale: 10 µm. **g**, x-y image (Z0 plane) and x-z image (Y0 plane) of z-stack images showing the U2OS cells expressing GFP-CaaX in vitronectin fibres, presented in **f** (right side). Scale: 10 µm. **h**, Zoom-in x-y images of the area indicated in **g** showing plasma membrane folding along vitronectin fibres at the middle (z = Z0), bottom (z = Z1), and top (z = Z2) of cells. This result suggests that cell plasma membranes are deformed by 3D ECM fibres. Scale: 10 µm.

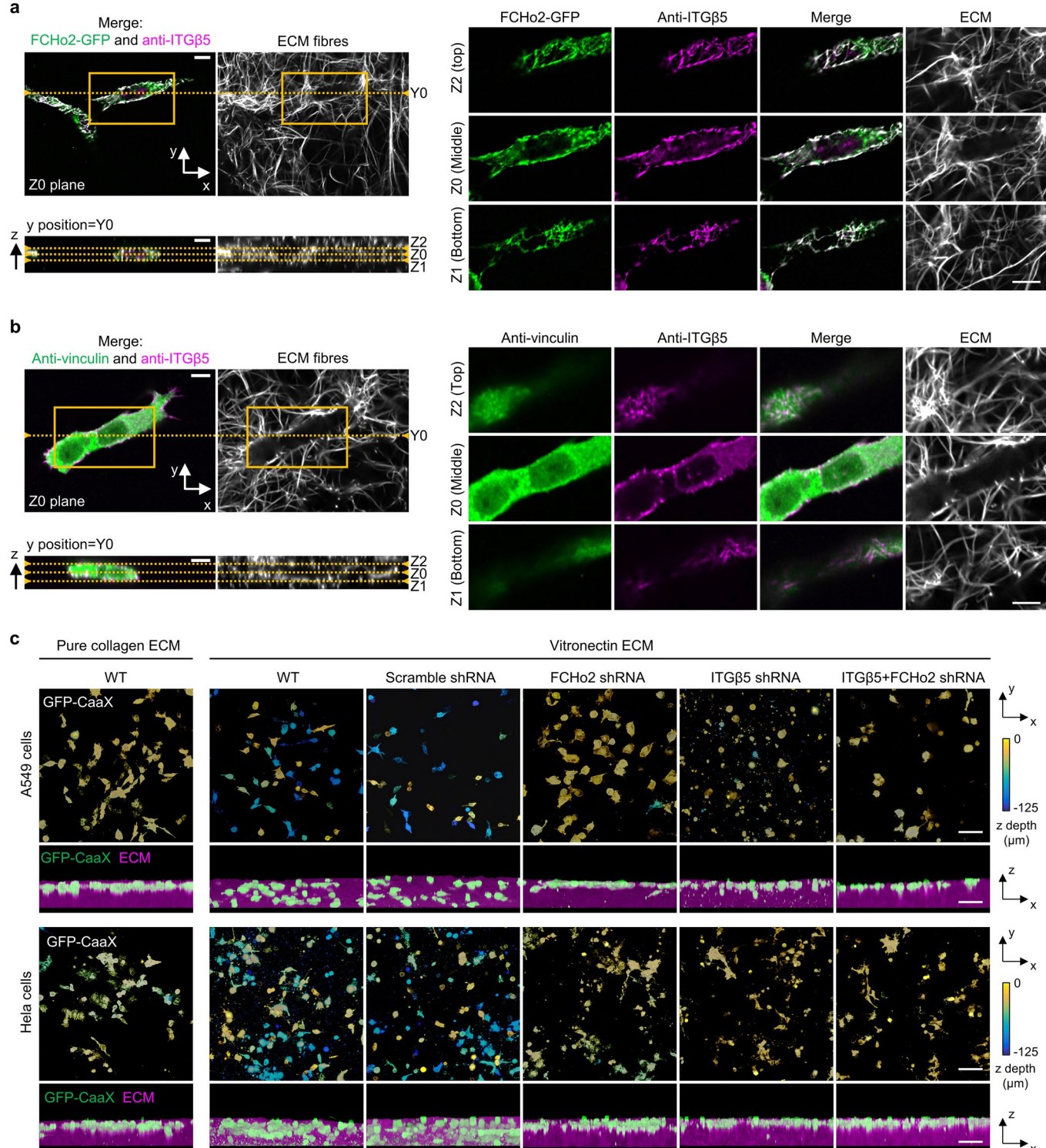

**Extended Data Fig. 10 | Curved adhesions in 3D ECMs facilitate cell migration in 3D ECMs. a**, x-y view (Z0 plane) and x-z view (Y0 plane) of z-stack images of immunolabelled ITGβ5 in U2OS cells expressing FCHo2-GFP. The cells are embedded in 3D ECM made of vitronectin fibres labelled with AF647-collagen. Zoom-in x-y images of the yellow-box area show abundant curved adhesions indicated by the colocalization of FCHo2 and ITGβ5. Curved adhesions form along vitronectin fibres at the middle (Z0 plane), bottom (Z1 plane), and top (Z2 plane) of cells. A zoom-in of the x-y slide at the Z0 plane has been shown in Fig. 5g (top). Scale: 10 μm. **b**, x-y view (Z0 plane) and x-z view (Y0 plane) of z-stack images of immunolabelled vinculin and ITGβ5 in U2OS. The cells are embedded in 3D ECM made of vitronectin fibres labelled with AF647-collagen. Zoom-in x-y images of the yellow-box area do not show clear focal adhesions indicated by the colocalization of vinculin and ITGβ5. Examination of different imaging planes, the middle (Z0 plane), bottom (Z1 plane), and top (Z2 plane) of cells, shows that the colocalization of vinculin and ITGβ5 is sparse. A zoom-in of the x-y slide at the Z0 plane has been shown in Fig. 5g (bottom). Scale: 10 μm. **c**, Representative 3D images of A549 cells (top) and HeLa cells (bottom) in 3D matrices after 72-hr culture. Cells were transduced to express GFP-CaaX via lentiviral infection. In the x-y projections, cells are colour-coded according to the z depth. In the x-z projections, cells are coloured in green and merged with the ECM (magenta). Cells can infiltrate into 3D matrices of vitronectin fibres, but not into 3D matrices of pure collagen fibres. The shRNA knockdown of FCHo2, ITGβ5, or both significantly inhibit cell infiltrations into 3D ECMs. Scale: 100 μm.

# Reporting Summary

## Statistics

For all statistical analyses, confirm that the following items are present in the figure legend, table legend, main text, or Methods section.

| n/a | Confirmed | |
|---|---|---|
| ☐ | ☒ | The exact sample size (*n*) for each experimental group/condition, given as a discrete number and unit of measurement |
| ☐ | ☒ | A statement on whether measurements were taken from distinct samples or whether the same sample was measured repeatedly |
| ☐ | ☒ | The statistical test(s) used AND whether they are one- or two-sided<br>*Only common tests should be described solely by name; describe more complex techniques in the Methods section.* |
| ☐ | ☒ | A description of all covariates tested |
| ☐ | ☒ | A description of any assumptions or corrections, such as tests of normality and adjustment for multiple comparisons |
| ☐ | ☒ | A full description of the statistical parameters including central tendency (e.g. means) or other basic estimates (e.g. regression coefficient) AND variation (e.g. standard deviation) or associated estimates of uncertainty (e.g. confidence intervals) |
| ☐ | ☒ | For null hypothesis testing, the test statistic (e.g. *F*, *t*, *r*) with confidence intervals, effect sizes, degrees of freedom and *P* value noted<br>*Give P values as exact values whenever suitable.* |
| ☒ | ☐ | For Bayesian analysis, information on the choice of priors and Markov chain Monte Carlo settings |
| ☒ | ☐ | For hierarchical and complex designs, identification of the appropriate level for tests and full reporting of outcomes |
| ☐ | ☒ | Estimates of effect sizes (e.g. Cohen's *d*, Pearson's *r*), indicating how they were calculated |

*Our web collection on statistics for biologists contains articles on many of the points above.*

## Software and code

Policy information about availability of computer code

| Data collection | Nikon NIS-Elements AR (updated routinely by the manufacturer), MetaMorph 4.8.3, Leica LAS X (updated routinely by the manufacturer) |
|---|---|
| Data analysis | MATLAB 2023a and 2017a (MathWorks), ImageJ (Fiji 2.13), Prism 9.5.1 (GraphPad software), and custom MATLAB codes which have been deposited to github (Link: github.com/wzhang5publication/Data-analysis). |

For manuscripts utilizing custom algorithms or software that are central to the research but not yet described in published literature, software must be made available to editors and reviewers. We strongly encourage code deposition in a community repository (e.g. GitHub). See the Nature Portfolio guidelines for submitting code & software for further information.

## Data

Policy information about availability of data

All manuscripts must include a data availability statement. This statement should provide the following information, where applicable:
- Accession codes, unique identifiers, or web links for publicly available datasets
- A description of any restrictions on data availability
- For clinical datasets or third party data, please ensure that the statement adheres to our policy

Source data are provided with this paper. All other data supporting the findings of this study are available from the corresponding author upon reasonable request. Plasmids originally generated in this study and their sequence information are available at Addgene (IDs: 205090, 205091, 205092, 205093, 205094 and 205095). The expression level of integrins in U2OS cells was obtained from the Human Protein Atlas project's RNA HPA cell line gene data available from v23.0.proteinatlas.org.

# Field-specific reporting

Please select the one below that is the best fit for your research. If you are not sure, read the appropriate sections before making your selection.

☒ Life sciences          ☐ Behavioural & social sciences          ☐ Ecological, evolutionary & environmental sciences

For a reference copy of the document with all sections, see nature.com/documents/nr-reporting-summary-flat.pdf

# Life sciences study design

All studies must disclose on these points even when the disclosure is negative.

| | |
|---|---|
| Sample size | No sample size calculation was performed. Sample size was determined based on our previous experimental experience (e.g., Zhao, W. et al. Nat. Nanotechnol. 12, 750–756 (2017); Lou, H.-Y. et al. PNAS 116, 23143–23151 (2019)). Similar sample sizes have been used in many recent papers from other labs, such as Shiu, J.-Y. et al. Nat. Cell Biol. 20, 262–271 (2018); Changede, R. et al. Nat. Mater. 18, 1366–1375 (2019); Oria, R. et al. Nature 552, 219–224 (2017). |
| Data exclusions | No data were excluded for data analysis. |
| Replication | All experiments were repeated independently at least two times with similar results. |
| Randomization | The experiments were not randomized. |
| Blinding | Blinding was not performed, as the protein sub-cellular distribution would clearly reveal the sample identity to investigators during most experiments. All data were collected automatically by instruments other than by human evaluation. |

# Reporting for specific materials, systems and methods

We require information from authors about some types of materials, experimental systems and methods used in many studies. Here, indicate whether each material, system or method listed is relevant to your study. If you are not sure if a list item applies to your research, read the appropriate section before selecting a response.

### Materials & experimental systems

| n/a | Involved in the study |
|---|---|
| ☐ | ☒ Antibodies |
| ☐ | ☒ Eukaryotic cell lines |
| ☒ | ☐ Palaeontology and archaeology |
| ☒ | ☐ Animals and other organisms |
| ☒ | ☐ Human research participants |
| ☒ | ☐ Clinical data |
| ☒ | ☐ Dual use research of concern |

### Methods

| n/a | Involved in the study |
|---|---|
| ☒ | ☐ ChIP-seq |
| ☒ | ☐ Flow cytometry |
| ☒ | ☐ MRI-based neuroimaging |

## Antibodies

| | |
|---|---|
| Antibodies used | Name Supplier Identifier Application Dilution<br>1.Rat anti-integrin β1 antibody (clone 9EG7) BD Biosciences Cat#550531 IF 1/100<br>2.Mouse anti-activated integrin β1 antibody (clone HUTS-4) Sigma-Aldrich Cat#MAB2079Z IF 1/100<br>3.Mouse anti-integrin αvβ5 antibody (clone P5H9) R&D Systems Cat#MAB2528 IF 1/100<br>4.Mouse anti-alpha adaptin antibody [AP6] Abcam Cat#ab2730 IF 1/250<br>5.Mouse anti-vinculin antibody (clone hVIN-1) Sigma-Aldrich Cat#V9131 IF 1/250<br>6.Mouse anti-paxillin antibody (clone 349/Paxillin) BD Biosciences Cat#610051 IF 1/250<br>7.Mouse anti-talin1 antibody (clone 8D4) Abcam Cat#ab157808 IF 1/100<br>8.Mouse Anti-Vitronectin/S-Protein antibody (clone VN58-1) Abcam Cat#ab13413 IF 1/250<br>9.Rabbit anti-FAK (phospho Y397) antibody (clone EP2160Y) Abcam Cat#ab81298 IF 1/250<br>10.Rabbit anti-Clathrin heavy chain antibody (Polyclonal) Abcam Cat#ab21679 IF 1/250<br>11.Rabbit anti-integrin β5 antibody (clone D24A5) Cell signaling technology Cat#3629S IF 1/250 and WB 1/1000<br>12.Rabbit anti-FCHo2 antibody (Polyclonal) Novus Biologicals Cat#NBP2-32694 WB 1/1000<br>13.Rabbit anti-GAPDH (clone 14C10) Cell signaling technology Cat#2118 WB 1/5000<br>14.Rabbit anti-GFP antibody (Polyclonal) Invitrogen Cat#A-11122 WB 1/1000<br>15.Mouse anti-mCherry antibody (clone GT857) Sigma-Aldrich Cat#SAB2702291 WB 1/1000<br>16.Alexa Fluor 488-conjugated goat anti-Mouse IgG (H+L) antibody Thermo Fisher Scientific Cat#A-11001 IF 1/500<br>17.Alexa Fluor 568-conjugated goat anti-Mouse IgG (H+L) antibody Thermo Fisher Scientific Cat#A-11004 IF 1/500<br>18.Alexa Fluor 488-conjugated goat anti-Rabbit IgG (H+L) antibody Thermo Fisher Scientific Cat#A-11034 IF 1/500<br>19.Texas Red-conjugated goat anti-Rabbit IgG (H+L) antibody Thermo Fisher Scientific Cat#T2767 IF 1/500<br>20.Alexa Fluor 647-conjugated goat anti-Rat IgG (H+L) antibody Thermo Fisher Scientific Cat#A-21247 IF 1/500 |

21.HRP-linked goat anti-Rabbit IgG (H+L) antibody Cell signaling technology Cat#7074 WB 1/1000
22.HRP-linked goat anti-Mouse IgG (H+L) antibody Cell signaling technology Cat#7076 WB 1/1000

Validation

The antibodies have been validated by the manufacturers and previous publications.
Details are available from the references below.
1.https://www.bdbiosciences.com/en-us/products/reagents/flow-cytometry-reagents/research-reagents/single-color-antibodies-ruo/purified-rat-anti-mouse-cd29.550531
2.https://www.sigmaaldrich.com/US/en/product/mm/mab2079z
3.https://www.rndsystems.com/products/human-integrin-alphavbeta5-antibody-p5h9_mab2528
4.https://www.abcam.com/products/primary-antibodies/alpha-adaptin-antibody-ap6-ab2730.html
5.https://www.sigmaaldrich.com/US/en/product/sigma/v9131
6.https://www.bdbiosciences.com/en-us/products/reagents/microscopy-imaging-reagents/immunofluorescence-reagents/purified-mouse-anti-paxillin.610051
7.https://www.abcam.com/products/primary-antibodies/talin-1-antibody-8d4-ab157808.html
8.https://www.citeab.com/antibodies/759488-ab13413-anti-vitronectin-s-protein-antibody-vn58-1
9.https://www.abcam.com/products/primary-antibodies/fak-phospho-y397-antibody-ep2160y-ab81298.html
10.https://www.abcam.com/products/primary-antibodies/clathrin-heavy-chain-antibody-ab21679.html
11.https://www.cellsignal.com/products/primary-antibodies/integrin-b5-d24a5-rabbit-mab/3629
12.https://www.novusbio.com/products/fcho2-antibody_nbp2-32694
13.https://www.cellsignal.com/products/primary-antibodies/gapdh-14c10-rabbit-mab/2118
14.https://www.thermofisher.com/antibody/product/GFP-Antibody-Polyclonal/A-11122
15.https://www.sigmaaldrich.com/US/en/product/sigma/sab2702291
16.https://www.thermofisher.com/antibody/product/Goat-anti-Mouse-IgG-H-L-Cross-Adsorbed-Secondary-Antibody-Polyclonal/A-11001
17.https://www.thermofisher.com/antibody/product/Goat-anti-Mouse-IgG-H-L-Cross-Adsorbed-Secondary-Antibody-Polyclonal/A-11004
18.https://www.thermofisher.com/antibody/product/Goat-anti-Rabbit-IgG-H-L-Highly-Cross-Adsorbed-Secondary-Antibody-Polyclonal/A-11034
19.https://www.thermofisher.com/antibody/product/Goat-anti-Rabbit-IgG-H-L-Cross-Adsorbed-Secondary-Antibody-Polyclonal/T-2767
20.https://www.thermofisher.com/antibody/product/Goat-anti-Rat-IgG-H-L-Cross-Adsorbed-Secondary-Antibody-Polyclonal/A-21247
21.https://www.cellsignal.com/products/secondary-antibodies/anti-rabbit-igg-hrp-linked-antibody/7074
22.https://www.cellsignal.com/products/secondary-antibodies/anti-mouse-igg-hrp-linked-antibody/7076

# Eukaryotic cell lines

Policy information about cell lines

Cell line source(s)

U2OS (ATCC HTB-96™), A549 (ATCC CCL-185™), HeLa (ATCC CCL-2™), Mouse Embryonic Fibroblast (MEF) (ATCC CRL-2991™), IMR-90 lung fibroblast cells (ATCC® CCL-186™, a gift from Scott Dixon), HEK293T cells (ATCC® CRL-3216™), HT-1080 (ATCC CCL-121™), U-251MG (Sigma-Aldrich, 09063001), human mesenchymal stem cells (PCS-500-011™), and MCF7 (ATCC HTB-22™)

Authentication

Cell line authentication was not performed. All cell lines were expanded from the original vials vendors provided and used for experiments within ten passages.

Mycoplasma contamination

All cell lines were tested negative for mycoplasma contamination by RT-PCR.

Commonly misidentified lines
(See ICLAC register)

We did not use commonly misidentified lines listed by ICLAC.

