## [Peer Review File · Nature Cell Biology]

Peer Review Information

Journal: Nature Cell Biology

Manuscript Title: Curved adhesions mediate cell attachment to soft matrix fibres in 3D

Corresponding author name(s): Professor Bianxiao Cui

Editorial Notes:

Transferred manuscripts This manuscript has been previously reviewed at another journal that is not operating a transparent peer review scheme. This document only contains reviewer comments, rebuttal and decision letters for versions considered at Nature Cell Biology.

Reviewer Comments & Decisions:

Decision Letter, initial version:
--

Our ref: NCB-A50756-T

6th June 2023

Dear Dr. Cui,

I am sorry once again for the delay, as Reviewer #2 was unable to re-review your revisions. We secured reviewer #1 to assess on your responses to Reviewer #2's previous concerns.

Thank you for submitting your revised manuscript "Curved adhesions mediate cell attachment to soft matrix fibres in 3D" (NCB-A50756-T). It has now been seen by the original referee(s) and their comments are below. The reviewers find that the paper has improved in revision, and therefore we'll be happy in principle to publish it in Nature Cell Biology, pending minor revisions to satisfy the referees' final requests and to comply with our editorial and formatting guidelines.

The current version of your manuscript is in a PDF format, so please email us a copy of the file in an editable format (Microsoft Word or LaTeX)-- we can not proceed with PDFs at this stage.

Please note that starting today through June 24th, I will be out of the office. I have cc'ed my brilliant colleague Dr. Sabrya Carim (sabrya.carim@springernature.com) who will gladly answer any questions you may have and help you in the next steps during my absence.

Thank you again for your interest in Nature Cell Biology Please do not hesitate to contact us if you have any questions.

Sincerely,
Daryl

Daryl Jason Verzosa David, PhD

Senior Editor, Nature Cell Biology
Nature Portfolio

Heidelberger Platz 3, 14197 Berlin, Germany
Email: daryl.david@nature.com
ORCID: <https://orcid.org/0000-0002-9253-4805>

Reviewer #1 (Remarks to the Author):

I went carefully through the questions that reviewer 2 had. I think the response of the authors is completely thorough and they meticulously addressed every aspect. The authors have a really strong case and I think NCB is the right journal for this work. I am convinced that this will be a highly cited paper. It is also very timely as the field of curvature sensing is more and more moving from endocytosis towards substrate interaction. Exciting work!

Decision Letter, final checks:

Our ref: NCB-A50756-T

19th June 2023

Dear Dr. Cui,

Thank you for your patience as we've prepared the guidelines for final submission of your Nature Cell Biology manuscript, "Curved adhesions mediate cell attachment to soft matrix fibres in 3D" (NCB-A50756-T). Please carefully follow the step-by-step instructions provided in the attached file, and add a response in each row of the table to indicate the changes that you have made. Please also check and comment on any additional marked-up edits we have proposed within the text. Ensuring that each point is addressed will help to ensure that your revised manuscript can be swiftly handed over to our production team.

We would like to start working on your revised paper, with all of the requested files and forms, as

soon as possible (preferably within two weeks). Please get in contact with us if you anticipate delays.

In recognition of the time and expertise our reviewers provide to Nature Cell Biology's editorial process, we would like to formally acknowledge their contribution to the external peer review of your manuscript entitled "Curved adhesions mediate cell attachment to soft matrix fibres in 3D". For those reviewers who give their assent, we will be publishing their names alongside the published article.

Nature Cell Biology offers a Transparent Peer Review option for new original research manuscripts submitted after December 1st, 2019. As part of this initiative, we encourage our authors to support increased transparency into the peer review process by agreeing to have the reviewer comments, author rebuttal letters, and editorial decision letters published as a Supplementary item. When you submit your final files please clearly state in your cover letter whether or not you would like to participate in this initiative. Please note that failure to state your preference will result in delays in accepting your manuscript for publication.

Cover suggestions

As you prepare your final files we encourage you to consider whether you have any images or illustrations that may be appropriate for use on the cover of Nature Cell Biology.

Nature Cell Biology has now transitioned to a unified Rights Collection system which will allow our Author Services team to quickly and easily collect the rights and permissions required to publish your work. Approximately 10 days after your paper is formally accepted, you will receive an email in providing you with a link to complete the grant of rights. If your paper is eligible for Open Access, our Author Services team will also be in touch regarding any additional information that may be required to arrange payment for your article.

Please note that *Nature Cell Biology* is a Transformative Journal (TJ). Authors may publish their research with us through the traditional subscription access route or make their paper immediately open access through payment of an article-processing charge (APC). Authors will not be required to make a final decision about access to their article until it has been accepted. Find out more about Transformative Journals

Please use the following link for uploading these materials:
[Redacted]

Best regards,

Kendra Donahue
Staff
Nature Cell Biology

On behalf of

Daryl Jason Verzosa David, PhD

Senior Editor, Nature Cell Biology
Nature Portfolio

Heidelberger Platz 3, 14197 Berlin, Germany
Email: daryl.david@nature.com

ORCID: <https://orcid.org/0000-0002-9253-4805>

Reviewer #1:

Remarks to the Author:

I went carefully through the questions that reviewer 2 had. I think the response of the authors is completely through and they meticulously addressed every aspect. The authors have a really strong case and I think NCB is the right journal for this work. I am convinced that this will be a highly cited paper. It is also very timely as the field of curvature sensing is more and more moving from endocytosis towards substrate interaction. Exciting work!

Author Rebuttal, initial submissions:

Response to remaining reviewer comments

Reviewer #1:

Remarks to the Author:

I went carefully through the questions that reviewer 2 had. I think the response of the authors is completely through and they meticulously addressed every aspect. The authors have a really strong case and I think NCB is the right journal for this work. I am convinced that this will be a highly cited paper. It is also very timely as the field of curvature sensing is more and more moving from endocytosis towards substrate interaction. Exciting work!

Response:

We appreciate the reviewer's positive feedback. We are glad to hear that the reviewer found our response to the questions raised by Reviewer 2 to be comprehensive and meticulous. It is encouraging to note Reviewer #1's recognition of the timeliness of our research in the field of curvature sensing. Once again, we would like to express our gratitude for the reviewer's detailed and constructive comments in the previous rounds of review, which have helped us improve the quality of this study.

Final Decision Letter:

Dear Dr Cui,

I am pleased to inform you that your manuscript, "Curved adhesions mediate cell attachment to soft matrix fibres in 3D", has now been accepted for publication in Nature Cell Biology.

Please note that *Nature Cell Biology* is a Transformative Journal (TJ). Authors may publish their research with us through the traditional subscription access route or make their paper immediately open access through payment of an article-processing charge (APC). Authors will not be required to make a final decision about access to their article until it has been accepted. Find out more about Transformative Journals

If you have not already done so, we strongly recommend that you upload the step-by-step protocols used in this manuscript to the Protocol Exchange (www.nature.com/protocolexchange), an open online resource established by Nature Protocols that allows researchers to share their detailed experimental know-how. All uploaded protocols are made freely available, assigned DOIs for ease of citation and are fully searchable through nature.com. Protocols and Nature Portfolio journal papers in which they are used can be linked to one another, and this link is clearly and prominently visible in the online versions of both papers. Authors who performed the specific experiments can act as primary authors for the Protocol as they will be best placed to share the methodology details, but the Corresponding Author of the present research paper should be included as one of the authors. By uploading your Protocols to Protocol Exchange, you are enabling researchers to more readily reproduce or adapt the methodology you use, as well as increasing the visibility of your protocols and papers. You can also establish a dedicated page to collect your lab Protocols. Further information can be found at www.nature.com/protocolexchange/about

With kind regards,
Daryl

Daryl Jason Verzosa David, PhD

Senior Editor, Nature Cell Biology
Nature Portfolio

Heidelberger Platz 3, 14197 Berlin, Germany
Email: daryl.david@nature.com
ORCID: <https://orcid.org/0000-0002-9253-4805>